# A high-resolution 3D atlas of the spectrum of tuberculous and COVID-19 lung lesions

Gordon Wells[1] , Joel N Glasgow[2] , Kievershen Nargan[1], Kapongo Lumamba[1],
Rajhmun Madansein[3] , Kameel Maharaj[3], Leon Y Perumal[4], Malcolm Matthew[4] , Robert L Hunter[5],
Hayden Pacl[6] , Jacelyn E Peabody Lever[6] , Denise D Stanford[7,8] , Satinder P Singh[7,9] ,
Prachi Bajpai[10], Upender Manne[10] , Paul V Benson[10] , Steven M Rowe[7,8] , Stephan le Roux[11] ,
Alex Sigal[1] , Muofhe Tshibalanganda[12] , Carlyn Wells[13] , Anton du Plessis[12,14] ,
Mpumelelo Msimang[15], Threnesan Naidoo[1,15,16] & Adrie J C Steyn[1,2,17,*]

## Abstract

Our current understanding of the spectrum of TB and COVID-19 lesions in the human lung is limited by a reliance on low-resolution imaging platforms that cannot provide accurate 3D representations of lesion types within the context of the whole lung. To characterize TB and COVID-19 lesions in 3D, we applied micro/nanocomputed tomography to surgically resected, post-mortem, and paraffin-embedded human lung tissue. We define a spectrum of TB pathologies, including cavitary lesions, calcium deposits outside and inside necrotic granulomas and mycetomas, and vascular rearrangement. We identified an unusual spatial arrangement of vasculature within an entire COVID-19 lobe, and 3D segmentation of blood vessels revealed microangiopathy associated with hemorrhage. Notably, segmentation of pathological anomalies reveals hidden pathological structures that might otherwise be disregarded, demonstrating a powerful method to visualize pathologies in 3D in TB lung tissue and whole COVID-19 lobes. These findings provide unexpected new insight into the spatial organization of the spectrum of TB and COVID-19 lesions within the framework of the entire lung.

**Keywords** calcification; COVID-19; granuloma; thrombosis; tuberculosis

**Subject Categories** Methods & Resources; Microbiology, Virology & Host Pathogen Interaction; Respiratory System
See also: **A Agrawal & D Agrawal** (December 2022)

## Introduction

Since the discovery of *Mycobacterium tuberculosis* (*Mtb*) in the 1880s, histological methods have been the primary means of investigating tuberculosis (TB) pathophysiology at the cellular level (Rich, 1944; Canetti, 1955). This low-throughput approach provides excellent two-dimensional (2D) results for very small regions of interest but cannot contextualize larger TB lesions within the greater lung architecture. Given that the full spectrum of TB lesions is present in an active human TB lung, the limitations of conventional histology, especially the ability to examine only small areas, hinder more detailed examination of human TB pathophysiology. Notably, the complete spectrum of TB lesions has not yet been systematically defined. This major gap in our knowledge has been perpetuated by a reliance on animal models which do not fully replicate human

1 Africa Health Research Institute, University of KwaZulu-Natal, Durban, South Africa
2 Department of Microbiology, University of Alabama at Birmingham, Birmingham, AL, USA
3 Inkosi Albert Luthuli Central Hospital and University of KwaZulu-Natal, Durban, South Africa
4 Perumal & Partners Radiologists, Ahmed Al-Kadi Private Hospital, Durban, South Africa
5 Department of Pathology and Laboratory Medicine, University of Texas Health Sciences Center at Houston, Houston, TX, USA
6 Medical Scientist Training Program, University of Alabama at Birmingham, Birmingham, AL, USA
7 Department of Medicine, University of Alabama at Birmingham, Birmingham, AL, USA
8 Cystic Fibrosis Research Center, University of Alabama at Birmingham, Birmingham, AL, USA
9 Department of Radiology, University of Alabama at Birmingham, Birmingham, AL, USA
10 Department of Pathology, University of Alabama at Birmingham, Birmingham, AL, USA
11 Bruker Belgium NV, Kontich, Belgium
12 Research Group 3D Innovation, Physics Department, Stellenbosch University, Stellenbosch, South Africa
13 CT Scanner Facility, Central Analytical Facilities, Stellenbosch University, Stellenbosch, South Africa
14 Object Research Systems, Montreal, QC, Canada
15 Department of Anatomical Pathology, National Health Laboratory Service, Inkosi Albert Luthuli Central Hospital, Durban, South Africa
16 Department of Laboratory Medicine & Pathology, Walter Sisulu University, Eastern Cape, South Africa
17 Centers for AIDS Research and Free Radical Biology, University of Alabama at Birmingham, Birmingham, AL, USA
*Corresponding author. Tel: +27 31 5210611; E-mail: asteyn@uab.edu

pulmonary TB and a paucity of human TB lung tissues for study (Hunter, 2011, 2016). To aid diagnosis and identification of disease-specific lesions, high-resolution three-dimensional (3D) imaging techniques are needed to visualize the microarchitecture of pulmonary TB within the context of the whole lung. Establishing an atlas of the spectrum of TB lesions in 3D is expected to facilitate mapping of the pathophysiological changes in the human tuberculous lung, improve our understanding of how the localized immune response fails to control TB, and guide the design of more accurate diagnostics and novel interventions.

Tuberculosis has been the deadliest infectious disease for many years but was surpassed by COVID-19 in 2020 (Roberts, 2021). Our understanding of the range of lung pathologies associated with severe COVID-19 is at an early stage. Importantly, a renewed interest in autopsies has led to critical insights into the mechanisms of COVID-19 (Sperhake, 2020; Borczuk, 2021), and stands in stark contrast to the lack of modern postmortem TB studies. Similar to TB, clinical imaging modalities have also been applied to COVID-19 (Qin et al, 2020; Dhawan et al, 2021). High-resolution imaging of Mtb- and SARS-CoV-2-infected tissue and lesions could identify new markers for routine clinical imaging, complement standard histological analyses, and may provide an important resource for therapeutic intervention.

One approach to establish a 3D atlas of pathogen-specific pathophysiology is X-ray computed tomography (CT), an invaluable tool for nondestructive imaging in medical diagnosis (Lee et al, 2010; Marchiori et al, 2011; Yeh et al, 2012). Compared to medical CT, nano/micro-CT (n/μCT) uses higher energy X-rays and smaller rotations between sample and detector (< 1°) to generate much higher resolution (du Plessis et al, 2017; O'Sullivan et al, 2018). Imaging of diseased lung with μCT has been demonstrated in live mice (Ruscitti et al, 2017) and human lung tissue with TB (Wells et al, 2021), COVID-19 (Ackermann et al, 2020, 2022; Walsh et al, 2021), and COPD (Ikura et al, 2004; Boon et al, 2016; Thiboutot et al, 2019), although most studies examined small sections. Ackerman et al. reported hierarchical phase-contrast tomography of entire organs, including COVID-19 lungs, using the Extremely Brilliant Source of the European Synchrotron Radiation Facility (EBS-ESRF) (Ackermann et al, 2020, 2022). While this unique instrument allows highly detailed visualization of soft tissue, it is not available to most researchers. However, the suitability of μCT for examining the spectrum of TB or COVID-19 lesions and contextualizing microscopic findings within whole lobes has not been reported and represents a gap in our knowledge.

High-resolution computed X-ray tomography (HRCT) is used clinically to aid TB diagnosis (Bajaj & Tombach, 2017; Nakamoto et al, 2018) and can detect phenotypes of Mtb infection such as bronchiectasis, cavity formation, and tissue consolidation (Im et al, 1993), but at lower resolution than μCT (Sundaram et al, 2010; Yanagawa et al, 2010). Also, clinically relevant but of low resolution, low-energy X-ray or "soft" X-ray (STX) CT is used for soft tissue imaging, such as the detection of masses in mammography (Buzug, 2011). Soft tissues attenuate X-rays poorly compared to more electron-dense materials like metal or bone. Imaging quality can be improved by suffusing tissue with contrast agents with high atomic mass like iodine or heavy metals. These interact to different degrees with different tissues and cellular components, thus improving internal contrast with X-ray imaging (Pauwels et al, 2013; O'Sullivan et al, 2018).

The purpose of this study was to explore the ability of μCT imaging to augment traditional histopathology of human TB and COVID-19 lung with 3D context and radio-opacity, and to assess the potential of μCT to characterize the spectrum of TB and COVID-19 lesions. Here, we use routine clinical HCRT, STX, and experimental μ/nCT to present a 3D atlas of the spectrum of lesions in the human TB lung and assessed the capacity of μCT to image formalin-fixed, paraffin-embedded (FFPE) resected TB lung tissues. We also applied μCT with contrast staining to COVID-19 human postmortem whole lungs. We demonstrate the utility of μCT for direct visualization and precise description of the continuum of lesions in detail, providing unanticipated new knowledge that advances our understanding of pulmonary infections.

# Results

## Parallel imaging modalities applied to human pulmonary TB tissue

Patient and lung sample selection is described in Appendix Table S1. Initial examination of samples revealed gross architectural distortion, conspicuous upper lobe cavitation, bronchiectasis, lung shrinkage, and fibrosis (Appendix Figs S1A–C, S2C, and S5).

While various modalities of computed tomography have been used clinically to appraise TB (Ankrah et al, 2016; Esmail et al, 2016; Malherbe et al, 2016), the suitability of these for examining TB lung tissue ex vivo remains unknown. Therefore, we assessed the ability of HRCT, STX, and μCT to image human pulmonary TB ex vivo (Fig 1A–D). We examined a tissue sample with pathological features consistent with TB, including cavitation and calcification (Fig 1E, Appendix Fig S1A). STX revealed lesions (Fig 1F) as denser regions. These lesions were also revealed by post-processing of HRCT (Fig 1G and H) with preset instrument settings (presets) intended for clinical diagnosis. Highly electron-dense areas in Fig 1G and H coincide with tubercles, calcifications, and necrotic lesions in the histopathological analysis (Fig 1I–M). The densest areas correspond to white nodules in the formalin-fixed tissue (Fig 1E) some of which were confirmed to be calcification (Fig 1K and M).

We examined a larger sample with two cavities and two mycetomas (Fig 1N, Appendix Fig S1B and C). μCT of contrast-enhanced tissue yielded high-resolution images allowing detailed examination of pathology (Fig 1O, Appendix Fig S3A) compared to HRCT and STX. STX revealed fewer electron dense areas and some internal structures at low resolution (Fig 1P and Q, and Appendix Fig S3B). Post-processing of HRCT scans revealed internal differences (Figs 1R–U and S3D and E). Both the "Onco-Thorax" (Fig 1R and S) and "Bone-Metal" (Fig 1T and U) presets aided visualization of diseased tissue. The "Onco-Thorax" preset performed better at revealing diseased areas with initial staining (Fig 1R), while imaging with the "Bone-metal" preset improved after destaining (Fig 1U).

In sum, routine hospital imaging provides meaningful insight into TB diseased tissue ex vivo. STX is slightly superior to HRCT for examining TB lung tissue. Both modalities can identify macroscopic features such as calcification, tubercles, necrosis, and cavitation. Clinical CT resolution is typically fixed, while the ability of the μCT instrument used here to yield high-resolution images is dependent on the proximity of the sample to the X-ray source.

## µCT of uninvolved TB lung tissue

To create a reference for describing diseased tissue, we used µCT to characterize contrast-stained uninvolved, healthy lung tissue from a lung cancer patient. We identified septa, airways, and vasculature (Fig 2A). Blood vessels absorbed the most iodine resulting in "bright" regions, while epithelial tissue is of intermediate intensity, and bronchioles, interlobular septa, and alveoli are "dark". µCT slices could be matched with corresponding histology (Fig 2A and B) to confirm identification of microanatomical features. 3D segmentation allowed identification of inter/intralobular septa and small airways (Fig 2A and C). In Fig 2A, the septa appear similar to bronchioles but can be distinguished by stepping through the µCT scan or 3D segmentation (Fig 2C).

We also used µCT to examine uninvolved tissue from a tuberculous lung (Fig 2D–L, Appendix Fig S2C). µCT without contrast staining revealed little internal structure. Contrast staining with alcohol soluble eosin was attempted but resulted in poor penetration (Fig S2A and B). However, contrast staining enabled identification of typical lung features (Appendix Fig S2D). Greater detail and higher resolution (4.1 µm voxel size) were obtained with nCT (Fig 2E) compared to µCT (Fig 2D). Airways adjacent to arteries could be visualized and segmented (Fig 2D–F, Movie EV1) as in the healthy control sample. However, septa were not visible by n/µCT or histology (confirmed with an adjacent sample; Appendix Fig S2E). In contrast to uninvolved, healthy tissue, segmentation of blood in uninvolved TB lung tissue revealed disconnected regions and dilated alveoli (Fig 2F) compared to normal lung (Fig 2C). Identification of alveoli, bronchi, and vasculature in the uninvolved, healthy tissue (Fig 2G and H) was confirmed by H&E (Fig 2I and J) and Masson's Trichrome (MT) staining (Fig 2K and L). These results show that µCT can accurately detect microscopic features in healthy tissue and uninvolved human TB lung tissue and complements conventional histology via alignment of virtual tissue slices with histological images.

## µCT of a TB cavity and pulmonary mycetoma

Prolonged TB can cause pulmonary cavitation, creating an ideal environment for secondary infections such as fungal growth (mycetomas). Cavitation and bronchiectasis dilate the airways predisposing them to Rasmussen's aneurysms that fail to thrombose, resulting in clinically evident hemoptysis (Seedat & Seedat, 2018). We applied µCT to image cavitation and fungal infection in a TB patient. Chest X-ray (Fig 3A) and HRCT (Fig 3B) revealed a lobulated mycetoma in a large multi-loculated interconnected cavity (Fig 3C and D). We used µCT to examine a sample containing two large mycetomas (designated M) and two large cavities (designated $C_1$, $C_2$; Fig 3E). µCT and 3D segmentation (Fig 3E and F) revealed the lobulated mycetoma in significantly greater detail, confirming that the two separate mycetomas are connected (Fig 3E), which would likely be missed during visual or histological appraisal. Contrast staining revealed complex folds within the mycetoma and an ill-defined boundary with the surrounding tissue (Fig 3F) that requires manual segmentation (Appendix Fig S4).

Further characterization revealed an absence of vasculature in the region of cavity wall thickening between the larger cavity ($C_2$) and the mycetoma (Fig 3E, Movie EV2). This was confirmed by H&E and MT histopathology of serial slices (Fig 3G–K), and which further revealed significant fibrosis (Fig 3J and K). Segmentation of blood revealed a high degree of hemorrhage (blood lakes, designated L) and disconnected vasculature around portions of the cavities and the mycetoma compared to healthier tissue (Fig 3E). In summary, µCT is capable of macroscopic and microscopic characterization of the range of TB disease, including cavitation, mycetomas, and vascular rearrangement.

## µCT of calcification and unstained TB lesions in paraffin-embedded blocks

A vast number of archived FFPE TB lung tissue specimens are available worldwide. Recent studies have shown that FFPE samples can be analyzed using µCT (Scott et al, 2015; Katsamenis et al, 2019). We used µCT to scan TB lung tissue in FFPE blocks before and after paraffin removal, without contrast staining.

We examined a sample containing visible calcium deposits and a cavity containing a mycetoma (Fig 4A, Appendix Fig S5A). Scanning at 20.0 µm voxel size revealed a limited number of internal features due to the relatively poor internal contrast (Fig 4B). Although calcification interferes with internal contrast, blood vessels and internal structure within the mycetoma were confirmed by histology (Fig 4C–E). Improved contrast and a higher resolution (12.0 µm voxel size) was obtained by paraffin removal and exclusion of calcium nodules from the field of view. This allowed contrast adjustments that revealed internal structures within the mycetoma, and evidence of hemorrhage (Fig 4F). Calcium nodules are readily apparent in FFPE samples (Appendix Fig S5B) and visible in µCT raw images (Fig 4G and H). Maximum intensity projection reveals calcium crystals and features such as blood vessels and mycetomas (Fig 4I). There was little difference in radio-opacity between soft tissue and surrounding paraffin compared to calcium (Fig 4J). Large calcium nodules were observed in the lung tissue, whereas numerous smaller nodules were observed inside the mycetoma (Fig 4I, K and L, Movie EV3). 3D segmentation allowed reconstruction of their distribution and shapes (Fig 4L), including ovoid, branched conglomerates, and needle-shaped structures (Appendix Fig S6). The intensity histogram (Fig 4M) reveals a distinct separation between soft tissue and large calcium nodules. Scanning of large calcium nodules at 8.5 µm voxel size revealed internal structures and lacunae (Fig 4N and O, Movie EV4).

To assess the capacity of µCT to characterize necrotic TB lesions in FFPE blocks, we scanned an FFPE block (Fig 5A) containing necrotic TB lesions as confirmed by H&E staining (Fig 5B). Necrotic lesions were easily discernible (Fig 5C), including gradients of increasing electron density within some lesions consistent with calcification (Fig 5D and E). Notably, we observed what are likely remnants of airways within these lesions (Fig 5D and F), which may point to the evolution of necrotic granuloma formation. Importantly, there was enough internal contrast to allow for segmentation of necrotic areas from surrounding tissue (Fig 5G).

From these observations, we conclude that µCT is useful for characterizing calcium deposits directly in FFPE tissues by providing details of internal structures, distribution, shapes, and sizes of calcium nodules. Further, removal of paraffin from FFPE tissues allows collection of more detailed microanatomical information.

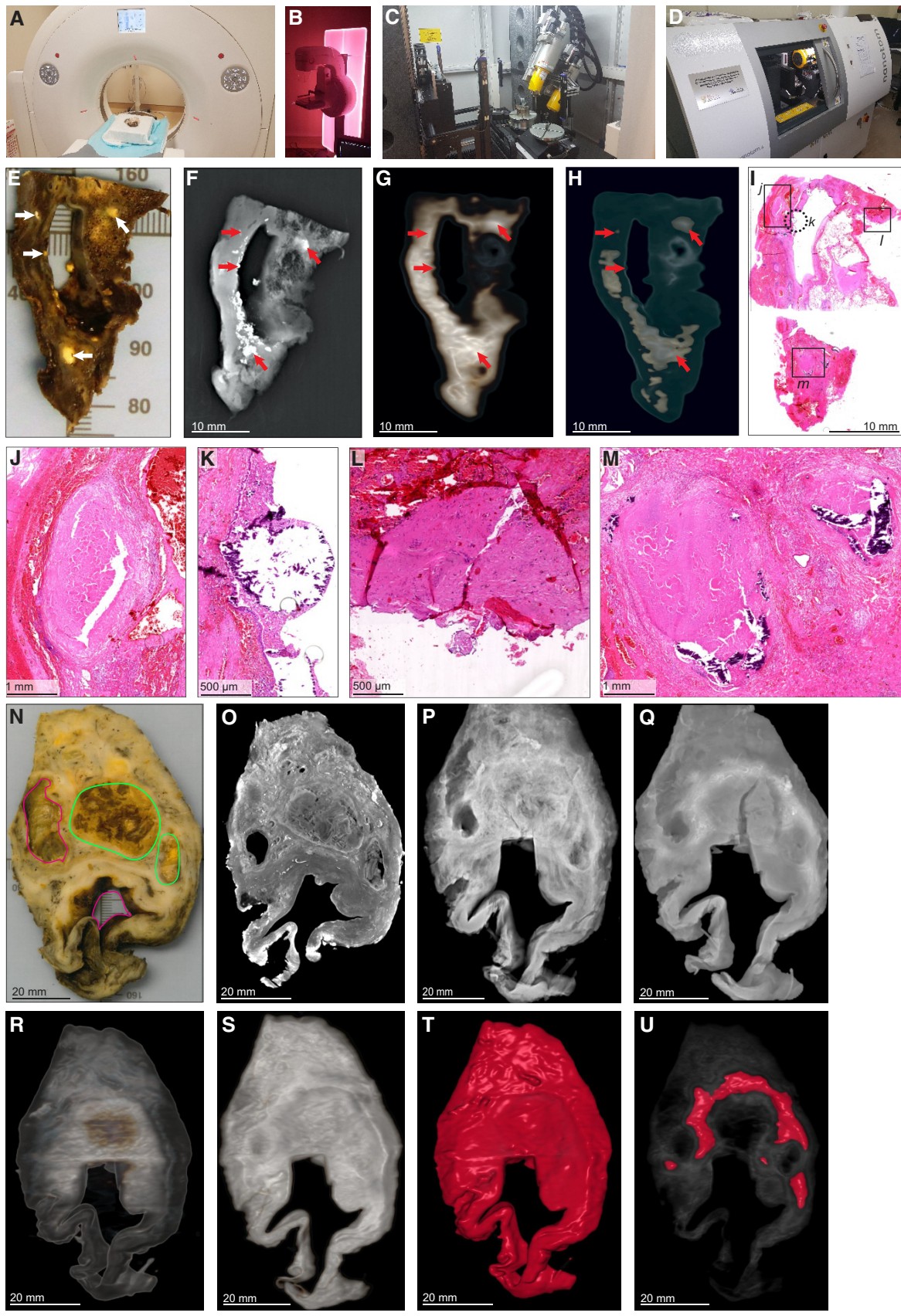

**Figure 1.**

**Figure 1.   Comparison of HRCT, STX, and µCT imaging of human TB lung.**

A       HRCT scanner: Siemens Somatom Perspective (64-slice).
B       Mammography STX scanner: Siemens Mammomat Inspiration System.
C       µCT scanner: General Electric Phoenix V|tome|x L240.
D       nCT scanner: General Electric Phoenix Nanotom S.
E–M    Imaging of Sample A exhibiting caseous necrosis and cavitation without contrast staining. (E) Macro-image. Necrotic and calcified legions are indicated by white arrows. (F) STX scan. Brighter regions correspond, in part, with regions of caseous necrosis that can be observed on the surface. Necrotic and calcified legions are indicated by red arrows. (G, H) Post-processed HRCT scans. Volumes of higher density can also be detected beneath the sample surface, and certain visualization presets within the HRCT software designed to aid clinical diagnosis can aid visualization of TB pathology. These visualizations, (G) the "Onco-thorax" preset, and (H) the "Onco-liver" preset, correspond, in part, with surface lesions and presumably with other denser tissue below the sample surface. Necrotic and calcified legions are indicated by the red arrows. (I) H&E histology of (E) (composite of two standard slides). (J–M) Close-ups of boxed lesions in (I) corresponding to regions of high-density visible in HRCT and STX. Calcification is indicated by dark-blue/purple lakes in (K) and (M).
N–U    Imaging of a sample B exhibiting cavitation, mycetoma, and caseous lesions from a sample with iodine contrast staining (Lugol's solution). The missing portion in panels P–U was excised for histology prior to scanning with clinical CT. (N) Gross image, with cavitation (purple) and mycetomas (green). (O) µCT slice of (N) at 60.0 µm voxel size. (P, Q) STX scan, before and after destaining, respectively. A small section (also observed in R–T) was removed from the cavity wall for histopathology. (R, S) HRCT with "Onco-thorax" preset post-processing before and after destaining, respectively. (T, U) HRCT with "Bone-metal" preset post-processing before and after destaining, respectively.

Lastly, we demonstrate that µCT can characterize necrotic and partially calcified necrotic TB granulomas in FFPE tissue.

### µCT of unattributed pathological abnormalities

Here, we set out to characterize the relationship between radio-opacity and histological features within necrotic granulomas. We scanned a lung specimen containing multiple radiological anomalies within necrotic lesions (Fig 6A, Appendix Fig S7) and compared this to corresponding H&E sections (Fig 6B). Figure 6C shows the rendering of the necrotic lesion in the context of surrounding tissue. Segmentation revealed that these undefined structures are largely encapsulated within the lesions (Fig 6D and E, Movie EV5) and formed branching, tube-like structures, consistent with obliterated airways (Fig 6F and G). Consistent with our previous report (Wells *et al*, 2021), we observed that granulomas can form complex, branched structures and display remarkable 3D heterogeneity (Fig 6C, E, H, and J) which would be difficult to appreciate by conventional histology. Compared to the common depiction of granulomas as spherical, this complex morphology may have profound implications for the treatment of TB (Wells *et al*, 2021).

Lastly, we combined scans of a sample before and after staining with iodine to evaluate "multi-channel" µCT. This allowed us to overlay calcium crystals relative to necrotic lesions (Fig 6I and J,

Appendix Fig S8). In one case, a calcium nodule was almost entirely embedded within a necrotic lesion, suggestive of the early onset of healing (Fig 6J and K).

These results demonstrate that µCT uniquely complements histo-pathological analysis by revealing hidden pathological structures that might otherwise be disregarded. These previously hidden structures strongly resemble obliterated airways, suggesting that *Mtb* undergoes bronchial dissemination consistent with previous studies (Wells *et al*, 2021). Lastly, these findings are likely to positively impact the discovery of previously unknown microanatomical structures in noninfectious diseases.

### µCT of SARS-CoV-2-infected human lung

To test the ability of µCT to evaluate other pulmonary pathologies, we examined postmortem lungs from SARS-CoV-2-infected decedents from South Africa and the USA (Appendix Table S1). Histology revealed organizing diffuse alveolar damage (Appendix Fig S9A), Type II pneumocyte hyperplasia with occasional squamous or bronchiolar metaplasia, intra-alveolar macrophages, and fibroblast proliferation as well as scattered cellular and acellular hyaline membranes. The presence of SARS-CoV-2 genomes and genomic replication intermediates was confirmed using *in situ* hybridization (Appendix Fig S9B). The right upper lobe from a

**Figure 2.   µCT of healthy and uninvolved human lung tissue.**

A–C    µCT and histological analysis of healthy lung, which was resected adjacent to cancerous lung, with iodine contrast staining (Sample C). (A) µCT image at 4.5 µm voxel size (top and side view), revealing inter/intralobular septa (green outline, blue arrow), alveoli (red arrows), airways (yellow arrows), and blood (green arrows). The side-view and surface (right image) were generated using 'ScatterHQ' rendering in MyVGL. (B) H&E histology of region depicted in (A), side view. Septa (blue arrow), alveoli (red arrow), airways (yellow arrow), and blood (green arrow). (C) 3D segmentation of blood (red), septa (green, partial segmentation), and bronchi (yellow).
D       µCT image at 20.0 µm voxel size (Sample D). Blood vessels (green arrows) and a bronchiole (yellow arrow).
E       nCT image at 4.1 µm voxel size. Blood vessels (green arrows) and a bronchiole (yellow arrow).
F       3D segmentation of (E) revealing vascular network (red) and airways (yellow). See also Appendix Fig S1.
G       µCT image of Sample D at 16.0 µm voxel size.
H       Magnified view of blood vessels (green arrows) and a bronchiole (yellow arrow) from the boxed area in (G).
I        Low power magnification of the H&E stain near to (G).
J        High power magnification of the H&E stain showing blood vessels (green arrows) and a bronchiole (yellow arrow) from the boxed area in (I).
K       Low power magnification of Masson's Trichrome (MT) stain.
L       High power magnification of MT stain showing blood vessels (green arrows) and a bronchiole (yellow arrow) from the boxed area in (K). MT stain produces red keratin and muscle fibers, blue/green collagen, red/pink cytoplasm, and brown/black nuclei.

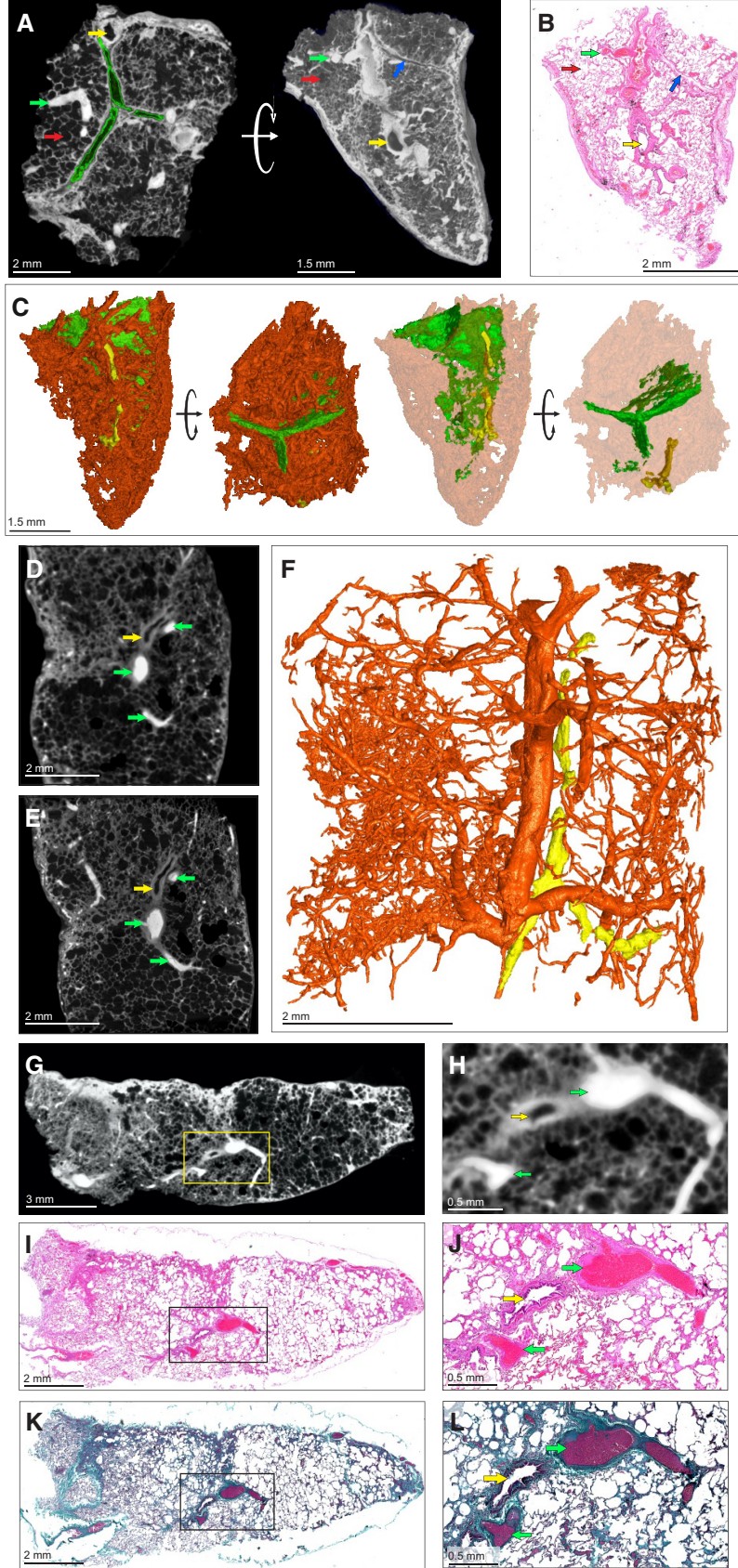

**Figure 2.**

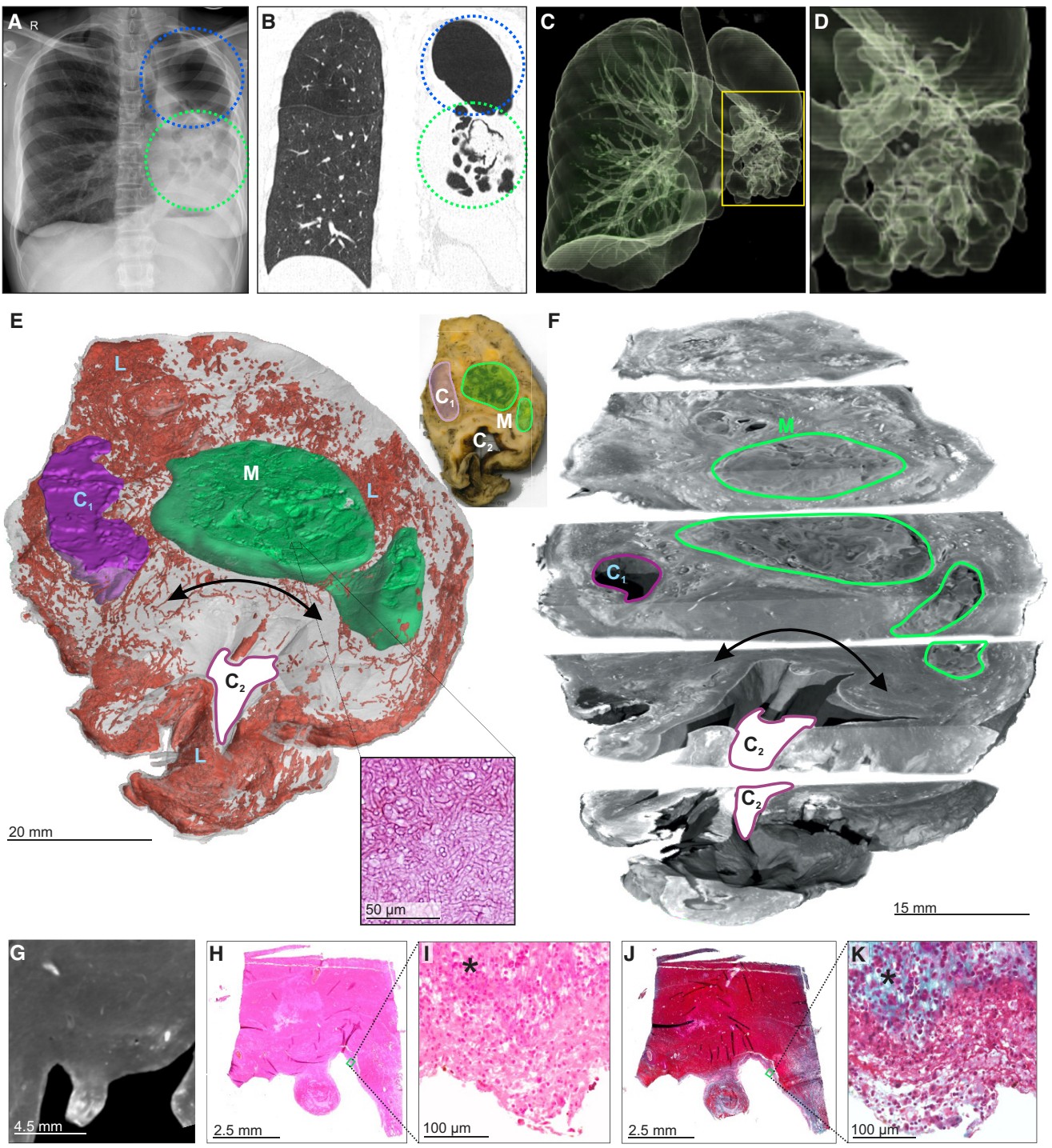

**Figure 3. µCT and histology of a lung resection sample with cavity and mycetoma.**

A  Chest X-ray of a TB patient exhibiting cavitation (blue circle) and mycetomas (green circle) in the left lung.

B  HRCT slice of same lung as (A) with cavitation (blue circle) and mycetoma (green circle).

C, D  3D reconstruction of HRCT from (B). (D) Magnified image of mycetoma.

E  3D segmentation of blood (red, 1279.12 mm$^3$), mycetoma (green, 3,709.62 mm$^3$, H&E histological confirmation of hyphae in boxed area), and partial cavity (purple, 673.13 mm$^3$), total sample volume: 26,739.92 mm$^3$. Top-right: macro-photo indicating the cavities (purple, "$C_n$") and mycetoma (green, "M"). Wall thickening around the cavity corresponds with a lack of vasculature (black double arrow). Hemorrhage is visible in the form of blood lakes ("L").

F  ScatterHQ (MyVGL) rendering of internal structure at $\pm$ 2 cm intervals and 5 mm below the top surface. The mycetoma is outlined in green ("M") and the upper cavity ("$C_1$"). The lower cavity ($C_2$) wall is indicated by the black double arrow. See also Movie EV2.

G  µCT close-up of cavity wall with strongly staining ridge.

H–K  H&E (H, I) and MT (J, K) histology of cavity wall with ridge reveals muscle (*) at medium and high magnification.

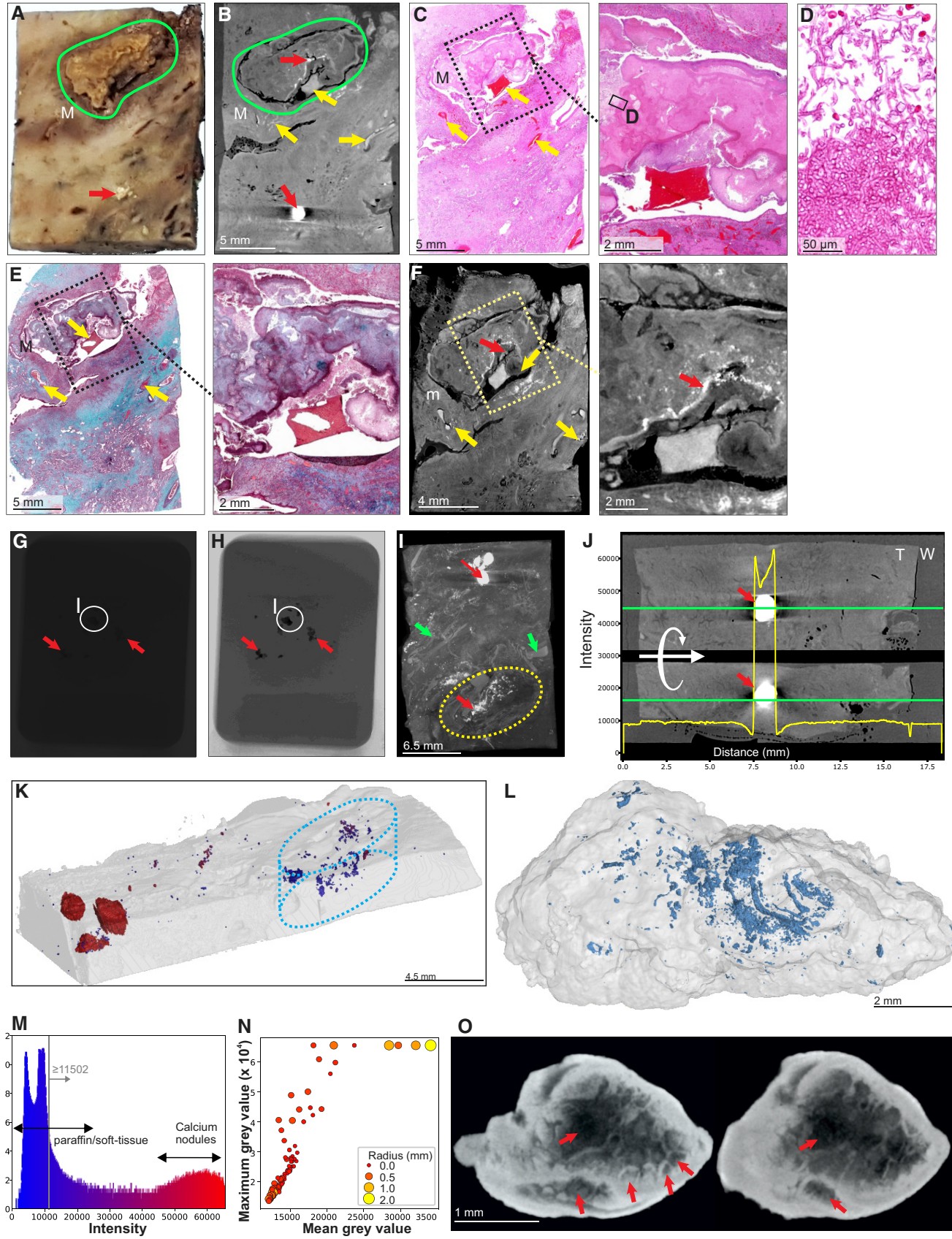

**Figure 4.**

**Figure 4. µCT and histology of FFPE samples, exhibiting calcification and mycetoma formation.**

A   Macrophotograph of Sample H, exhibiting calcification (red arrow) and a mycetoma (green outline, "M").

B   µCT slice of (A) at 26.8 µm voxel size. Determination of internal structure is limited compared to contrast staining with iodine. Nonetheless, calcification (red arrows) and to a lesser extent vasculature (yellow arrows) can be visualized.

C   H&E histology of (A) with medium power image of the mycetoma. Yellow arrows indicate blood vessels.

D   High power histology image of boxed area from (C) confirms the presence of hyphae in the mycetoma.

E   MT histology of (A) with close-up of the mycetoma cavity. Yellow arrows indicate blood vessels.

F   µCT of after paraffin removal (12.0 µm voxel size) and excluding calcified regions results in improved internal contrast with close-up of the mycetoma cavity. Yellow arrows: blood vessels, red arrow calcification.

G   Example of a single raw image of µCT scan with some calcifications indicated (red arrows; Sample I).

H   Edited version of (G); use of Corel PhotoPaint reveals calcium deposits (dark spots, red arrows).

I   Maximum Intensity Projection of reconstructed volume of Sample H, revealing calcification (red arrows), blood vessels (green arrows), and mycetoma (yellow circle).

J   Typical opacity profile (yellow graph) measured along the green axis in unstained paraffin block across a calcium deposit. Paraffin wax; "W", tissue; "T"; calcium; red arrow. The similar intensity of paraffin and soft tissue hindered differentiation within the tissue compared to scanning after paraffin removal. Calcium deposits are easy to distinguish; however, they do give rise to artifacts around the crystal (red arrow).

K   Segmentation of high-density regions in (I) (Sample H). Regions of interest are above the intensity threshold of 11,502 (grey line, see panel M) and are colored according to their maximum intensity within each region of interest (ROI).

L   Segmentation of high-density volumes (likely lower density calcification) within the mycetoma (dotted oval regions: I, K) in Sample H shown in Fig 4K and Appendix Fig S5A. See also Movie EV3.

M   X-ray intensity histogram for µCT of Fig 4K. The grey line approximately represents the threshold separating paraffin from soft tissue. The right bulge (red) corresponds to calcification, while the intermediate region corresponds to lower levels of calcification within soft tissue.

N   µCT can be used to quantify calcification in TB-infected tissue: Mean vs Max grey values of ROI from (K) with volume > 30 voxels. Circle size and color corresponds to ROI radius of gyration. High intensities were clamped (leveling of max values at 65,535) in order to improve contrast within soft tissue regions.

O   nCT slices (8.0 µm voxel size) of a calcium nodule reveals internal heterogeneity and lacunae (red arrows). See also Movie EV3.

Source data are available online for this figure.

severely diseased decedent was investigated in more detail (Fig 7A) and sectioned into ~ 1-cm-thick slices (Fig 7B) and scanned (Fig 7C, Appendix Fig S10A–F). The entire lower lobe was also removed and scanned (Fig 7D). Maximum intensity projection revealed an increased prevalence of dilated and tortuous peripheral vasculature compared to the central region (Fig 7C and D). Peripheral vessels could be clearly observed with 3D segmentation of a slice from the upper lobe (Fig 8A and B). Within the periphery, microvascular hemorrhage showed characteristic white patchy areas (Fig 8C). 3D segmentation revealed hemorrhage ranging from focal to extensive (Fig 8B and C). These findings are reminiscent of the "vascular tree-in-bud" sign identified in COVID-19 patients caused by hypercoagulability and lack of fibrinolysis, suggested as an indicator of pulmonary thrombotic angiopathy (Patel *et al*, 2020). In contrast, chest CT angiography (CTA) analysis demonstrated poor resolution of the vascular tree-in-bud sign (Fig 8D). Also, CTA analysis from two COVID-19 decedents identified irregularly dilated small peripheral pulmonary arteries with abnormal non-dichotomous branching pattern (Fig 8E), albeit at poor resolution. Compared to CTA (Fig 8F), the resolution of µCT made it possible to rapidly identify and characterize tortuous blood vessels (Figs 7D and 8G).

Lung slices from three of four COVID-19 patients revealed a predominance of vasculature near the periphery (Appendix Fig S11). Consistent with the known pathology of COVID-19, we observed occlusive organizing thromboemboli present in both small and large pulmonary arteries using histopathology and µCT (Appendix Fig S12A–D). We examined maximum intensity scans of the upper lobe (Fig 7C, Movie EV6) and identified multiple thrombi (Appendix Figs S12B and E, S13, and S14) including an occlusive thrombus at the juncture of two blood vessels (Appendix Fig S12B) and an adherent, partially occlusive thrombus in a pulmonary artery (Appendix Fig S12C–E). Importantly, whereas histopathological appraisal identified a spherical

occlusion (Appendix Fig S12C and D), µCT and 3D segmentation revealed that the occlusion is an organizing, cylindrical thrombus with early re-canalization at the site of adherence to the vessel wall (Appendix Fig S12E).

Importantly, we could match H&E histology with µCT scans (Appendix Fig S13). For example, gross dilated alveolar spaces (Appendix Fig S13A) and hemorrhage (Appendix Fig S13B) were confirmed with matched histology. This is consistent with studies demonstrating that thrombotic and micro-angiopathic pathology contribute to organ failure and mortality (Fox *et al*, 2020). Compared to the prominent peripheral vasculature observed in several COVID-19 lungs (Fig 7C and D, Appendix Fig S11), the non-COVID-19 control lungs lack the prominent peripheral vasculature (Appendix Fig S15).

In sum, we demonstrate that µCT is well suited for 3D reconstruction of COVID-19 vascular angiopathy in remarkable detail in whole lung lobes. Analysis of COVID-19 lobes using µCT and contrast staining is more accessible than the EBS-ESRF (15, 16) and represents an important benchmark for future routine study of lung diseases. Validating µCT with histopathology may further inform our understanding of how thrombotic microangiopathy contributes to COVID-19 pathology. A notable finding was the detail of dense peripheral vascular arrangement compared to the central region of the lung. These findings extend beyond what is typically seen using histopathology or HRCT and provide new insight into the pathophysiology of COVID-19.

## Discussion

Histology yields detailed 2D information within small areas of interest, but it cannot contextualize pulmonary lesions within the greater lung architecture. Here, we demonstrate the utility of clinical CT, STX, and µ/nCT for 3D analysis of FF and FFPE TB and

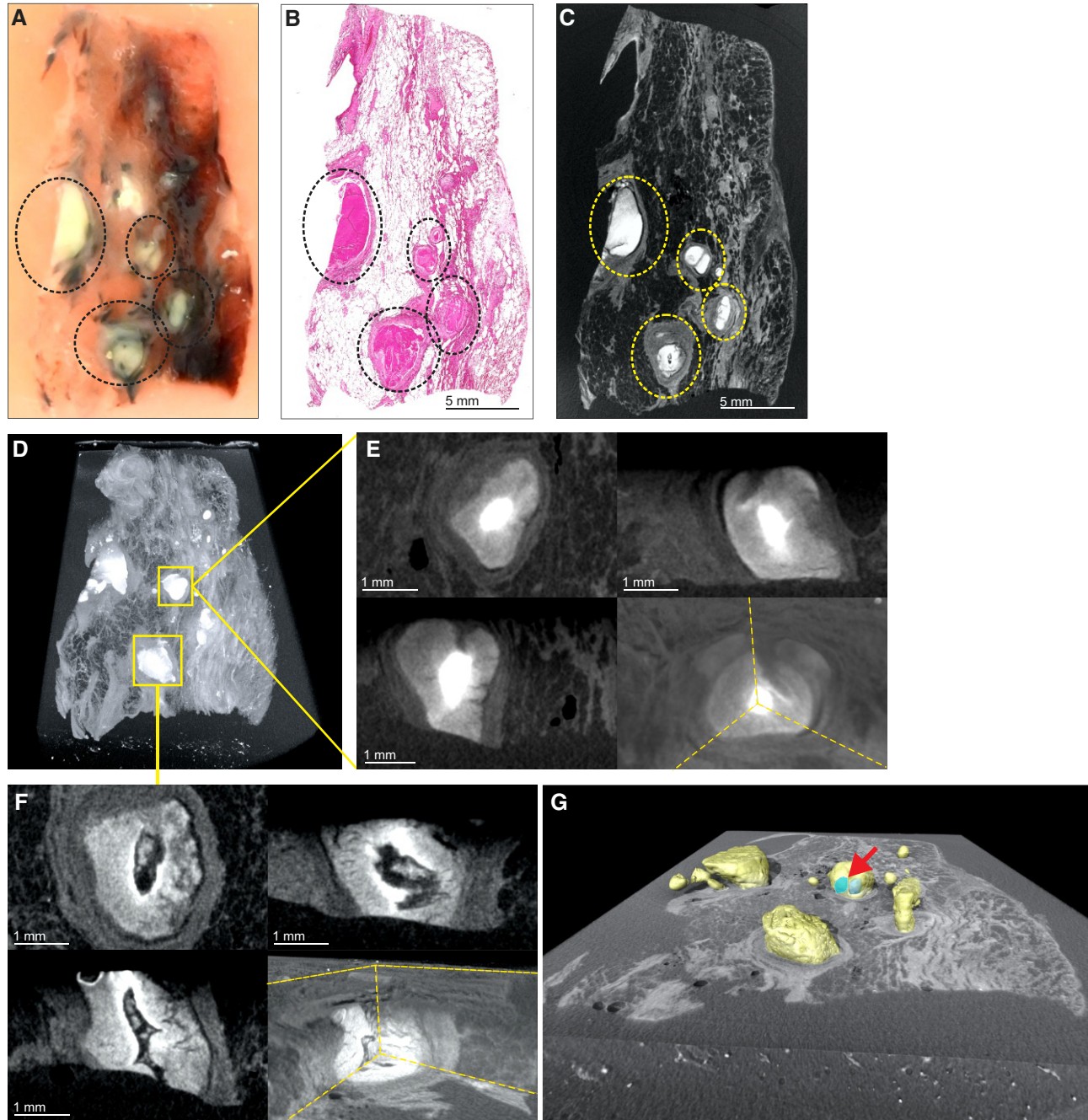

**Figure 5. µCT of FFPE necrotic granulomas.**

A  Image of necrotic TB lesions in a FFPE block. Dotted line circles: areas containing necrotic lesions.
B  H&E section of (A). Dotted line circles: areas containing necrotic lesions.
C  µCT section of (A). Dotted line circles: areas containing necrotic lesions.
D  Maximum intensity projection of µCT volume.
E  High power images of necrotic lesion with embedded calcification. Notice the gradient of electron density.
F  High power image of a necrotic lesion with airway remnants.
G  3D segmentation of necrotic lesion (yellow), calcification (blue, red arrow) embedded in a clipped view of X-ray attenuation.

postmortem COVID-19 lung tissue *ex vivo*. To the best of our knowledge, this is the first study to use µ/nCT to elucidate macro- and microscopic features of TB lesions such as cavitation, calcification, and necrosis, as well as COVID-19 pathophysiology across large lung samples in 3D. A major new contribution of this study is the characterization of obliterated airways in TB and hemorrhage from

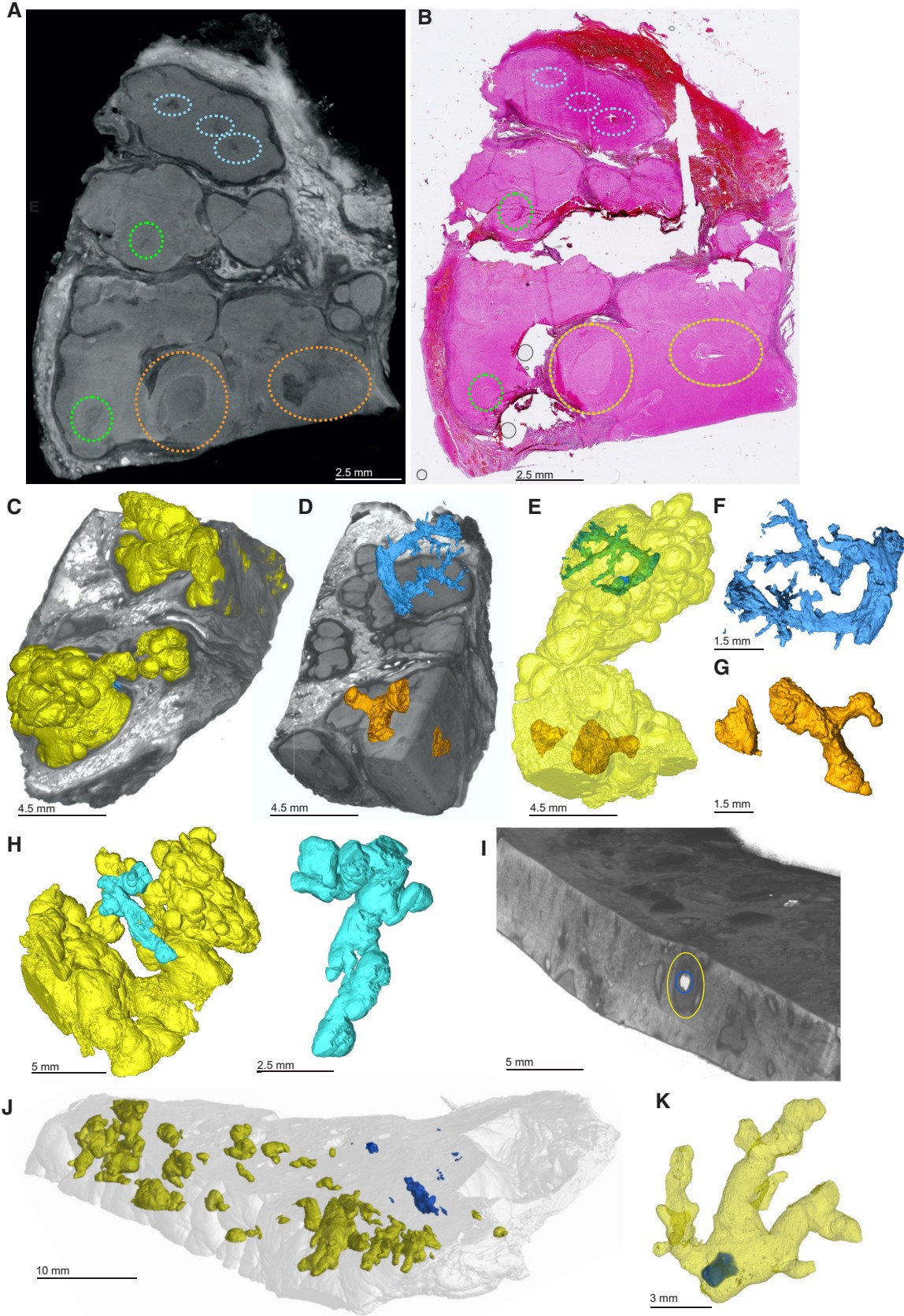

**Figure 6.**

**Figure 6. μCT characterization of pathological anomalies within necrotic granulomas.**

A  μCT slice of human TB lung specimen exhibiting caseous necrotic granulomas. Regions circled in blue, orange, and green contain pathological anomalies not readily identifiable in the corresponding histological analysis (B) but are amenable to manual segmentation (C–G). Not all abnormalities were segmented (e.g., green circles). Further adjacent slices are displayed in Appendix Fig S7.

B  H&E histology of (A) with regions containing anomalies circled as in (A).

C–G  Segmentation of unidentified branched structure in the dotted blue and orange circled regions from (A). (C) Rendered granulomas (yellow volumes) in relation to clipped electron density rendering. (D) Branched volumes in relation to clipped electron density rendering of necrotic granulomas. (E) Branched volumes in relation to rendered lesion surfaces (yellow). (F, G) Isolated rendering of branched volume in blue ovals and orange-circle regions from (A), which corresponds to obliterated airways.

H  Alternate views of independent granulomas (yellow, cyan) to demonstrate the spatial arrangement and complexity of granuloma morphology.

I  Overlay of scans with and without iodine contrast staining demonstrating the presence of a calcium crystal (blue) within a necrotic lesion (circled in yellow).

J  3D rendering overlay of necrotic lesions (yellow) and calcium crystals (blue) across entire sample.

K  3D rendering of calcium crystal (blue) within a necrotic lesion (yellow).

ruptured blood vessels in COVID-19 lungs that would not be possible with conventional 2D platforms. Further, μCT analysis of an entire COVID-19 lung lobe in 3D represents a technical advance that enabled us to contextualize vascular pathology within the greater lung architecture and visualize vascular micro-architecture in remarkable detail. A key finding was the dense peripheral vasculature compared to the central region of the COVID-19 lung. Our findings exemplify how 3D pathophysiology can extend beyond standard histopathology to improve our understanding of TB and COVID-19 disease. Overall, creating an atlas of TB and COVID-19 pulmonary lesion types establishes a foundation for a more accurate understanding of the pathogenesis of pulmonary pathogens, and is anticipated to help guide the development of innovative diagnostics and therapies.

There are few μCT studies that describe bacteria- or virus-infected human lungs. We recently used μCT to reveal the unusual heterogeneity of necrotic lesions in the human TB lung (Wells et al, 2021). Synchrotron-based CT has been used to study COVID-19 lung pathology at the alveolar and organ level (Ackermann et al, 2020, 2022; Walsh et al, 2021). Since histopathological analysis of an entire lobe or lung is not possible, μCT is an attractive platform for studying lung pathophysiology in large samples ex vivo. We identified an unusual spatial arrangement of vasculature within whole COVID-19 lobes (Figs 7C and D, and 8), which is consistent with pulmonary thrombotic microangiopathy that can lead to deep vein thromboses and large pulmonary thromboemboli. Detailed 3D segmentation of blood vessels revealed microangiopathy associated with hemorrhage, previously shown to contribute to death of patients with severe COVID-19 (Fox et al, 2020).

Our data illustrate that μ/nCT is a powerful imaging tool that can contribute to a more informative clinicopathological analysis of TB and COVID-19 sequelae. Firstly, 3D segmentation enables the reconstruction of pathophysiological abnormalities that are not detectable via conventional 2D histology. For example, 3D segmentation of cryptic abnormalities within necrotic TB lesions revealed the remnants of obliterated airways (Fig 6). These findings are consistent with other postmortem CT studies (Im et al, 1993), and provide new insight into the evolution of TB lesions, suggesting that the caseous material filling the bronchiolar lumen induces bronchial wall necrosis, which promotes progressive necrosis of the lesion. Secondly, the exceptional sensitivity of μ/nCT toward identifying calcified lesions has important implications for the study of TB latency since partially and fully calcified lesions are present in the lungs of decedents that died from causes other than TB (Opie &

Aronson, 1927; Aronson & Whitney, 1930; Sweany et al, 1943). Hence, μ/nCT could be used as a tool to screen postmortem lung tissue for calcified lesions, which could be further analyzed for the presence of viable, culturable, and virulent Mtb bacilli. Thirdly, hemorrhage in COVID-19 is routinely appraised via histopathology, but identification of the source, location, blood vessel size, and manner of rupture is difficult. μCT followed by 3D segmentation allows the identification of the structure, location, and features of ruptured blood vessels (Fig 8C), which may improve our understanding of the evolution of pulmonary infarction and hemorrhage in COVID-19. Fourthly, μCT analysis enabled us to identify and characterize thrombi and nonobstructive organized occlusions in COVID-19 tissue. For instance, whereas histopathological appraisal identified an apparently spherical thrombus, 3D segmentation accurately identified the structure as a cylindrical shape fused to the lumen of a blood vessel. Hence, 2D appraisal of this organized occlusion is markedly different from its corresponding 3D structure. Lastly, the combination of μCT imaging, histopathology, and tissue-clearing technologies like CLARITY and CUBIC (Chung & Deisseroth, 2013; Susaki et al, 2015) have the potential to broadly influence other disciplines including pathology, biomedical imaging, infectious diseases, and cancer by identifying previously unrecognized pathological abnormalities.

We also found that μCT is valuable for characterizing cavities. Cavitation occurs during the most contagious stage of TB disease and is the most difficult type of lesion to treat successfully (Canetti, 1955). While routine pathological inspection may indicate multiple cavities, 3D segmentation can determine whether cavities are interconnected. Cavities can be thin- or thick-walled and are important indicators of active disease. Cavities are generated within 24 hours to months (Canetti, 1955) when obstructive lobular pneumonia triggers necrosis and necrotic debris is expelled by coughing (Hunter, 2016, 2018). HRCT, and particularly μCT, can resolve small cavities that cannot be observed by standard chest X-ray (Im et al, 1993).

Mtb can trigger formation of calcium deposits that are mostly dystrophic, occurring in coagulative, caseous, and liquefactive necrotic tissue (Morrison, 1970). While calcification is not specific to Mtb infection, this process is indicative of healing although the mechanism(s) of deposition is poorly understood. HRCT can detect calcium deposits (typically ≥ 1 mm; Ohno et al, 2012), but μCT can detect much smaller deposits. Interestingly, 3D segmentation allowed characterization of the internal structure of calcium deposits and revealed voids of unknown origin. μCT also revealed

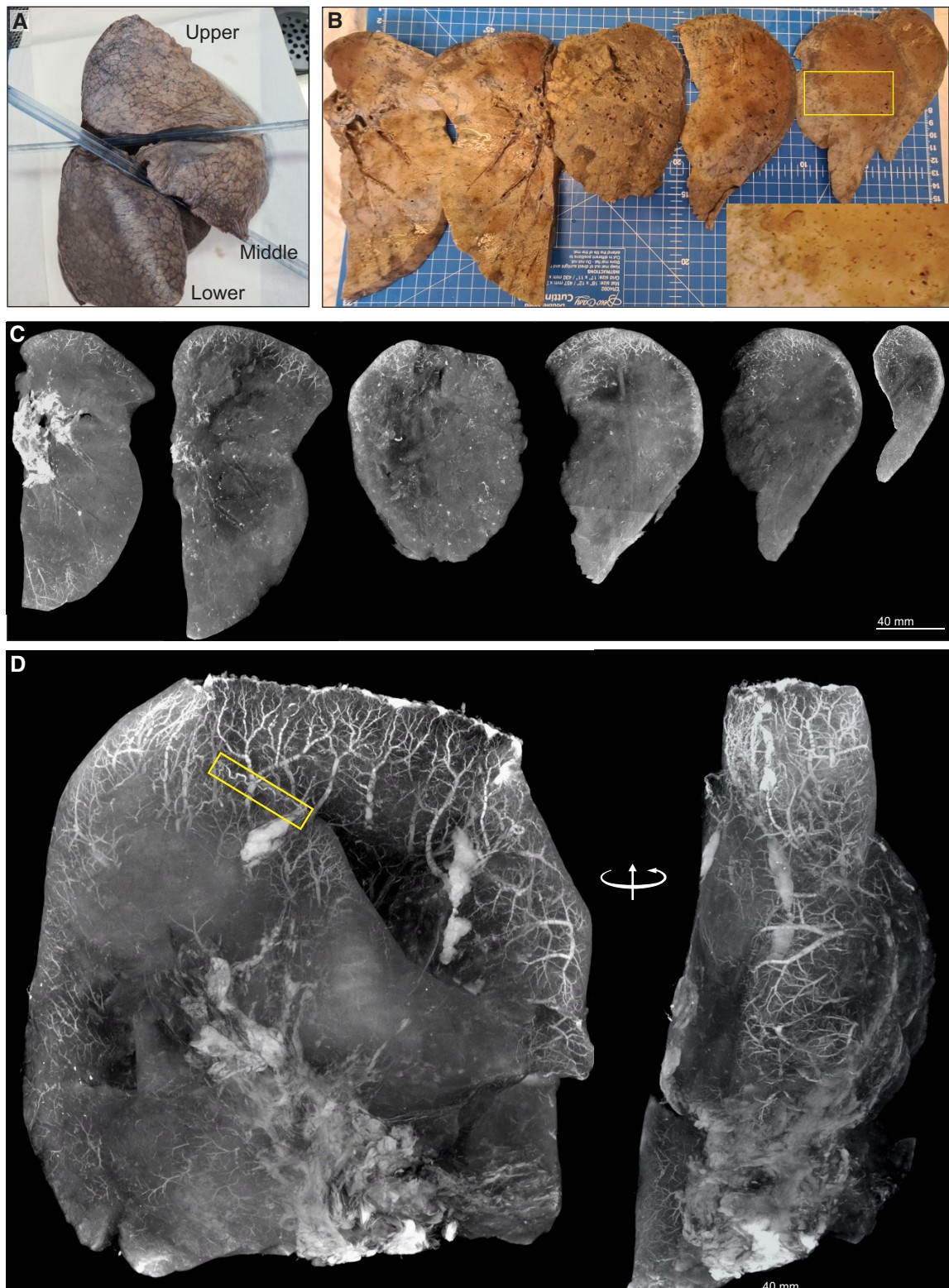

**Figure 7. Vascularization in COVID-19 lungs.**

A   Gross image of formalin-fixed right lung. The upper and lower lobes were selected for µCT.
B   Macrophotographs of lower lobe sections with medium power magnification (yellow box inset).
C   Maximum intensity projections of separate µCT scans for sections from (B).
D   Maximum intensity projection of lower lobe from (A). A tortuous blood vessel (yellow box) is indicated.

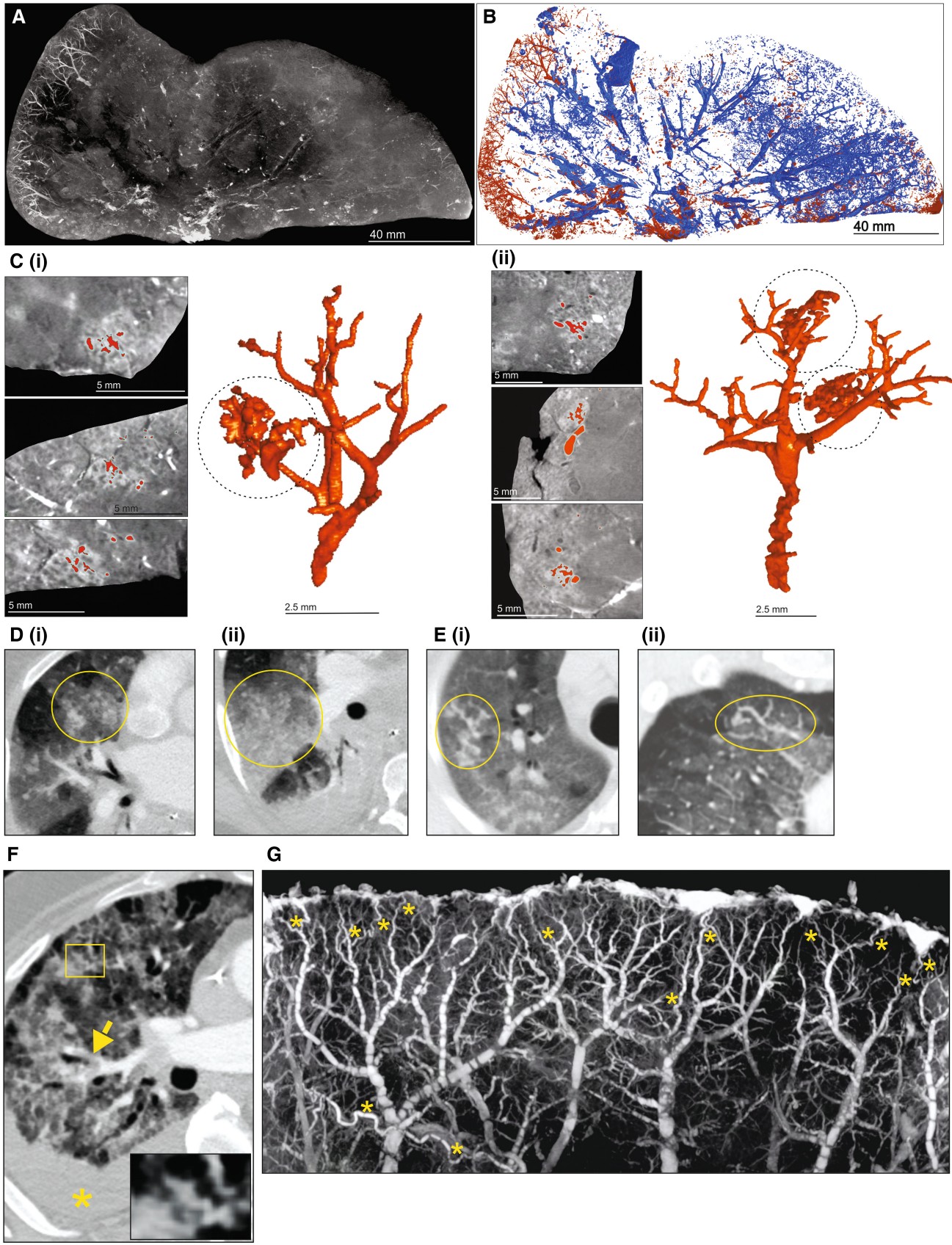

**Figure 8.**

**Figure 8. Vascularization and vascular abnormalities in COVID-19 lungs.**

A    μCT imaging of vascularization in a 1 cm-thick slice of right upper lobe of a COVID-19 lung.

B    Segmentation of blood (red) from and airways (blue) from (A).

C    Segmentation of blood and blood-pooling in tissue shown in (A) (red); (i), (ii). Two separate of examples of pooling. Left: μCT from three orthogonal views. Right: 3D segmentation.

D–F  Chest CTA shows (D) vascular tree-in-bud (yellow oval) appearance, (E) non-dichotomous branching (yellow oval) and (F) tortuous dilated blood vessel (yellow rectangle), pulmonary thromboembolism in small sub-segmental pulmonary arteries (yellow arrow), and ground glass opacities due to COVID-19 pneumonia and dependent pleural effusion (yellow asterisk).

the presence of calcium deposits in a mycetoma, which is unusual since these deposits are typically dystrophic. *Aspergillus fumigatus* and *A. niger* are the fungi most commonly responsible for this under-studied complication in chronic pulmonary TB, and colonize ~ 20% of cavities, especially in the upper lobes (Klein & Gamsu, 1980; Gefter, 1992). The identification and characterization of calcified TB lesions are important to the TB field as historical studies have shown that in latent TB postmortem tissue, these lesions frequently contain live and virulent *Mtb* (Opie & Aronson, 1927, Aronson & Whitney, 1930, Sweany *et al*, 1943).

Although there is a small difference in X-ray attenuation between paraffin and lung tissue, μCT allowed partial characterization of tissue in wax blocks. Contrast was improved by deparaffinization (which can be reversed), which may allow CT-based examination of archived FFPE specimens. Considering the vast number of FFPE samples available globally, scanning and annotation of these valuable tissue libraries may reveal new structural markers to complement histopathology.

Characterization of TB lung tissue *ex vivo* using routine clinical HRCT and mammographic imaging provided mixed results depending on sample type and the use of a contrast agent. Calcified deposits and macroscopic tubercles were identified by both methods; STX appears to be the superior modality and identified macroscopic lesions. However, STX is restricted to a rotation limit of 90° compared to 360° with HRCT. HRCT is more sensitive than chest radiography and can differentiate between active and inactive lesions (Hatipoglu *et al*, 1996; Lee *et al*, 2010). HRCT and STX have the potential to contribute to *ex vivo* imaging studies when μ/nCT is unavailable.

Limitations of this study include a limited number of heterogenous tissue samples, the use of different μCT scanners in two locations, and the use of a single contrast agent. In addition, similar to histopathology, shrinkage of tissue caused by formalin fixation and contrast staining is widely known, but unavoidable. Future studies could include a comprehensive atlas of latent, subclinical, and other stages of TB and COVID-19, including miliary and extrapulmonary TB.

In summary, our data demonstrate that μ/nCT is a powerful imaging tool for the near-histological level study of TB and COVID-19 sequelae. Several findings were made regarding the 3D spectrum of TB lesions, including the granuloma shape, calcium deposition, cavitation, and the identification of histologically unattributed pathophysiological features that were not detectable by conventional histopathology. In COVID-19 lungs, an unexpected finding was the disproportionately dense and dilated vascularization close to the pleural surface in whole lobes. We anticipate that μ/nCT could be used to establish a 3D reference atlas of the human tuberculous lung (including latent and subclinical TB) derived from digitized 3D image libraries of tissue, organs from new patients, and existing FFPE libraries. This atlas could be used to identify novel

imaging biomarkers based on patterns of differential radio-opacities (Waterhouse *et al*, 2019), and be implemented routinely as instrumentation becomes more accessible. Lastly, we expect that an atlas of the lesion types will inform our understanding of the failure of localized immunity and be an important resource for therapeutic and diagnostic development.

# Materials and Methods

### Ethics and human subjects

This study was approved by the University of KwaZulu-Natal Biomedical Research Ethics Committee (Class approval study number BCA 535/16). Patients undergoing lung resection for TB (Study ID: BE 019/13) were recruited from King DinuZulu Hospital Complex, a tertiary center for TB patients in Durban, South Africa. *Mtb*-infected human lung tissues are routinely obtained following surgery for removal of irreversibly damaged lobes or lungs (bronchiectasis and/or cavitary lung disease). UAB IRB approval was provided for tissue distribution of SARS-CoV-2 RNA in autopsy and surgical pathology cases (IRB-300006562) and UAB authorization for autopsy included consent for use of tissue for research. Routine clinical chest CTA with 2 × 2 mm slice thickness was performed on decedents with COVID-19 pneumonia on a Philips Brilliance CT 64 scanner. Written informed consent was obtained from all participants. All patients undergoing lung resection for TB had completed a full 6- to 9-month course of anti-TB treatment, or up to 2 years of treatment for drug-resistant TB. Patients were assessed for extent of cavitation and/or bronchiectasis via HRCT. Karnofsky scores, 6-min walk test, spirometry, and arterial blood gas were used to assess the fitness of each patient to withstand a thoracotomy and lung resection. Assessment of patients with massive hemoptysis included their general condition, effort tolerance prior to hemoptysis, arterial blood gas measurement, serum albumin level, and HRCT imaging of the chest. On gross assessment, all pneumonectomies or lobectomies were bronchiectatic, hemorrhagic, variably fibrotic, and atelectatic and contained visible tubercles (Appendix Table S1). Experiments using human samples conformed to the principles set out in the WMA Declaration of Helsinki and the Department of Health and Human Services Belmont Report. Due to the experimental nature of the μCT scanning platforms and severity of disease, no blinding was done.

### Patient recruitment and sample acquisition

Patients undergoing lung resection for TB were recruited from King DinuZulu Hospital Complex in Durban, South Africa. Typically, patients are referred when they experience recurrent chest

infections, recurrent minor hemoptysis, massive hemoptysis, failed therapy, infection with drug-resistant Mtb, empyema thoracis, or pneumothorax (Chinta *et al*, 2018; Reddy *et al*, 2018) (Appendix Table S1).

## μ/nCT scanning

A General Electric Phoenix V|tome|x L240 system (2,024 × 2,024 pixel image, 16-bit depth) was used for μCT (except Fig 5) with a voxel size range of 12.0–60.0 μm. A General Electric Phoenix Nanotom S (2,304 × 2,304 pixel image, 16-bit depth) was used for nCT with a isotropic voxel size range of 4.1–16.0 μm. Although the nCT instrument is capable of sub-micrometer resolution for small samples, the samples analyzed in this study were too large. All samples were scanned over 360°. A range of settings were used to scan the samples as described in Appendix Table S2. Briefly, voltage varied between 50 and 160 kV, current varied between 200 and 1,000 μA, and scanning times ranged from 2,000 to 5,400 s. For most scans, a tungsten target was employed. A molybdenum target was used for two nCT scans (Appendix Table S2).

One sample (Fig 5) was scanned in a Bruker SkyScan 2214 X-ray microscope. A total of 1,801 images were recorded with the flat-panel at 60 kV, 149 μA with an averaging of 5.

A MiLabs 3D/4D micro-resolution CT (μCT) imaging device (MILabs, Utrecht, Netherlands) with dynamic contrast-enhanced imaging using MILabs Acquisition Software (MILabs, Utrecht, Netherlands) was used to generate the μCT scans for samples in Appendix Figs S11 and S15. μCT acquisition parameters are set for Ultra-Focused magnification and Accurate scan mode. Additional settings are as follows: Tube Voltage (55 kV), Tube Current (0.19 mA), Scan Angle (360°), Angle Step Degree (0.75°), Binning (×22), Exposure (20 ms), Beam Hardening (0.971111), and HU Calibration (10449.444444 HU). MILabs Reconstruction software (MILabs, Utrecht, Netherlands) was used to process μCT scans. Reconstructed scans were converted with MIPAV v10 (Medical Image Processing, Analysis, and Visualization, https://mipav.cit.nih.gov/) to DICOM format for further processing in VGStudioMax.

## Sample preparation

Representative sections of cavitational and parenchymal abnormalities were selected for imaging. Tissue samples (Appendix Table S2) from resected human lungs (un-inflated; Appendix Table S1) were selected for HRCT, STX, or μ/nCT analyses. All samples were fixed in 10% buffered formalin for at least 14 days. Samples A and B were obtained from resected lungs with evidence of cavitation and Aspergillus infection in sample B. Samples C, D and G represent relatively healthy tissue from a cancerous lung and Mtb-infected lung, respectively. Sample F was selected from a lung with evidence of severe TB infection including extensive caseous necrosis. Samples H and I exhibit calcification, and fungal infection in H. Samples B, C, D, F, G, and K–U were contrast stained with iodine by immersing the samples in 2.5% Lugol's solution for 1–5 days depending on sample size. Samples G and H were also mounted in paraffin blocks before scanning. For μ/nCT scanning, samples were mounted in 50-ml falcon tubes, or in plastic containers with gaskets for the larger samples. The tube/container was then secured to the stage with cellophane tape and/or florist's foam. Non-paraffin-embedded samples were lodged above a formalin bath in the bottom of the tube or container with polystyrene foam and lodged between the walls of the tube or container to prevent shifting of the sample. The low density of polystyrene foam also enables easy deletion from the reconstructed volume during subsequent visualization and analysis. The tube was then sealed with parafilm for the duration of the scan to maintain a moist atmosphere and prevent desiccation. Prior to mounting, samples were rinsed with water and blotted dry to remove excess staining solution.

## Clinical CT scanning

A Siemens Somatom Perspective (64-slice) hospital scanner was used to generate HRCT scans for Samples A and B. The low energy ("soft") X-ray scans (STX) were performed with a Siemens Mammomat Inspiration System. Post-processing was performed with Siemens Syngo.via VB10B imaging software. Formalin-fixed lung tissue samples were scanned in plastic bags to prevent dehydration. Clinical evaluation of the STX and HRCT scans is performed with proprietary software (Syngo.via) supplied by the HRCT and STX manufacturer. This software contains default options or "presets" designed to highlight typical anatomical and pathogenic features in X-ray scans.

## Numerical analysis and plotting

Opacity plots, histograms, and scatterplots were generated using Python in the Jupyter notebook environment with the Matplotlib, Seaborn, and Pandas libraries.

## Image processing and volume rendering

Volumes were reconstructed with system-supplied General Electric Datos software, except for Fig 5 which was reconstructed in NRecon 2.1.0.1 (Bruker). Subsequent visualization and analysis (such as volume and density calculations) were performed in Volume Graphics VGStudio Max 3.1 or 3.2. Where possible, simple thresholding was employed for segmentation (demarcation of 3D regions of interest), followed by semi-automated segmentation using the VGStudio Max region growing tool. The region growing tool allows for manual selection of a 3D scan region based on adjustable intensity thresholds and different intensity averaging schemes. Two approaches were used for segmenting vasculature with the region growing tool. First, by using a stringent threshold and selecting a voxel near the center of a brightly stained vessel, it is possible to rapidly generate branched segmentations that do not overlap into non-vascular tissue. Second, individual vessels can be manually extended by selecting adjacent volumes within an overlapping sphere and careful adjustment of the thresholds for intensity values with smaller differences to non-vascular tissue. This latter mode was also used for segmenting necrotic lesions (Appendix Fig S4). For complex heterogeneous volumes, this mode is needed to segment intricate structures without including adjacent voxels that represent a different tissue. To correlate with histopathology, the axes of the 3D volume were adjusted (re-registered in VGStudio Max parlance) followed by slicing through the volume to match the 2D histology image as closely as possible.

## Histopathology

Identification of CT scan features was confirmed by histological techniques using Hematoxylin and Eosin (H&E) or Masson's trichrome (MT) stain. Briefly, samples of lung were fixed in 10% buffered formalin and processed in a vacuum filtration processor using a xylene-free method with isopropanol as the main substitute fixative. Tissue sections were embedded in paraffin. Sections were cut at 4 μm, baked at 60°C for 15 min, dewaxed through two changes of xylene and rehydrated through descending grades of alcohol to water. These sections were stained with H&E or the MT method using standard procedures. Slides were dehydrated in ascending grades of alcohol, cleared in xylene, and mounted with a mixture of distyrene, plasticizer, and xylene.

## Histology slide digitization and cross-validation with μ/nCT imaging

Human lung specimens were digitized using a Hamamatsu NDP slide scanner (Hamamatsu NanoZoomer RS2, Model C10730-12) and its viewing software (NDP.View2). The red, green, and blue color balance was kept at 100%, whereas gamma correction was maintained between 0.7 and 2. Brightness (60–110%) and contrast (100–180%) settings vary slightly between slides depending on staining quality. Resolution was 230 nm/pixel yielding a file size of 2–4.4 GB. Contrast, brightness, and intensity of exported images (jpg format) were minimally adjusted using CorelDraw X8. Registration of the μ/nCT scans against histopathology images was performed manually in VGStudio Max by using blood vessels, bronchi, and lesions as landmarks.

## RNAscope analysis

*In situ* hybridization (ISH) assays were performed using an RNAscope® kit and ISH probes according to the manufacturer's instructions (Advanced Cell Diagnostics, ACD, Newark, CA, USA). SARS-CoV-2 replication is indicated by the presence of a replicative RNA intermediate detected as a red signal (C2 channel, nCoV2019-orf1ab-sense-C2 [cat. no. 859151-C2]). SARS-CoV-2 viral load was detected using the RNA probe V-nCoV2019-S (cat. no. 848561) specific for the gene encoding the spike (S) glycoprotein, which was detected as a turquoise signal. The RNAscope® 2.5 Duplex Reagent Kit (cat. no 322430) along with Human (Hs) Positive Control Probes for housekeeping genes PPIB-C1/ POLR2A-C2, (cat. no 321641) were used to assess RNA integrity. Simultaneously, consecutive sections were probed with probes targeting dihydrodipicolinate reductase B mRNA of a *Bacillus subtilis* strain (DapB) as a negative control, (cat. no. 320751) to assess the specificity of the assay.

# Data availability

High resolution 5× scans and photographs of the histological data can be obtained from the European Bioinformatics Institute BioImage Archive under accession number S-BIAD516 (https://www.ebi.ac.uk/biostudies/BioImages/studies/S-BIAD516).

**Expanded View** for this article is available online.

---

**The paper explained**

**Problem**

Our current understanding of pulmonary diseases, such as tuberculosis (TB) and COVID-19, is limited by conventional histological methods based on 2D assessment of small regions of interest. To aid diagnosis and identification of disease-specific lesions, high-resolution three-dimensional (3D) imaging techniques are needed to visualize the microarchitecture of pulmonary TB and COVID-19 lesions within the context of the whole lung.

**Results**

We tested the ability of X-ray microscopy (also known as micro-Computed Tomography, μCT) to complement traditional histology and provide 3D imaging of lung pathology. Several findings were made regarding the 3D spectrum of TB lesions, including granuloma shape, calcium deposition, cavitation, and the identification of histologically unattributed pathophysiological features that were not detectable by conventional histology. In COVID-19 lungs, we observed disproportionately dense and dilated vasculature close to the pleural surface in whole lobes. Further, we successfully applied μCT to formalin-fixed tissue, including paraffin-embedded samples.

**Impact**

This study demonstrates that μCT is a powerful imaging tool for studying TB and COVID-19 pathology at near-histological levels in 3D. We anticipate that μCT could be used to establish a 3D reference atlas of the human tuberculous lung derived from digitized 3D image libraries of tissue, organs from new patients, and existing fixed-tissue libraries. This atlas could be used to identify novel imaging biomarkers based on patterns of differential radio-opacities. Lastly, we expect an atlas of TB and COVID-19 lesion types will inform our understanding of the failure of localized immunity and be an important resource for therapeutic and diagnostic development.

## Acknowledgements

This work was supported by NIH grants R01AI111940, R01AI134810, R01AI137043, R33AI138280, and R21A127182, Bill and Melinda Gates Foundation award OPP1130017, funding from the Wellcome Leap ΔTissue Program, and pilot funds from the UAB CFAR and UAB Heersink School of Medicine to AJCS. Additional support came from CRDF Global, the South African Medical Research Council, and a South African NRF BRICS Multilateral grant to AJCS. Other support included R35HL135816-04S1, P30DK072482, and U01HL152978 to SMR; 5F31HL146083-02 and 2T32HL105346-11A1 to JEPL and 3T32GM008361-30S1 to HP and JEPL. This work was also supported by the Cystic Fibrosis Foundation (ROWE19R0) to SMR, and a Bill and Melinda Gates Foundation Award OPP1116944 to AS.

## Author contributions

**Gordon Wells:** Conceptualization; formal analysis; funding acquisition; investigation; visualization; methodology; writing—original draft; writing—review and editing. **Joel N Glasgow:** Investigation; writing—original draft; project administration; writing—review and editing. **Kievershen Nargan:** Validation; investigation; methodology; writing—review and editing. **Kapongo Lumamba:** Data curation; investigation; visualization; writing—review and editing. **Rajhmun Madansein:** Validation; investigation; methodology; writing—review and editing. **Kameel Maharaj:** Formal analysis; investigation; writing—review and editing. **Leon Y Perumal:** Resources; investigation; methodology; writing—review and editing. **Malcolm Matthew:** Resources; formal analysis; investigation; methodology; writing—review and editing. **Robert L Hunter:** Validation; investigation; writing—review and editing.

**Hayden Pacl:** Investigation; visualization; writing—review and editing. **Jacelyn E Peabody Lever:** Resources; investigation; visualization; writing—review and editing. **Denisie D Stanford:** Resources; investigation; visualization; writing—review and editing. **Satinder P Singh:** Resources; investigation; methodology; writing—review and editing. **Prachi Bajpai:** Investigation; methodology; writing—review and editing. **Upender Manne:** Investigation; methodology; writing—review and editing. **Paul V Benson:** Resources; formal analysis; investigation; methodology; writing—review and editing. **Steven M Rowe:** Resources; investigation; methodology; writing—review and editing. **Stephan le Roux:** Investigation; visualization; methodology; writing—review and editing. **Alex Sigal:** Resources; writing—review and editing. **Muofhe Tshibalanganda:** Resources; investigation; visualization; writing—review and editing. **Carlyn Wells:** Investigation; visualization; methodology; writing—review and editing. **Anton Plessis:** Resources; investigation; visualization; methodology; writing—review and editing. **Mpumelelo Msimang:** Resources; formal analysis; investigation; writing—review and editing. **Threnesan Naidoo:** Conceptualization; resources; formal analysis; supervision; validation; investigation; methodology; writing—original draft; writing—review and editing. **Adrie J C Steyn:** Conceptualization; resources; supervision; funding acquisition; investigation; methodology; writing—original draft; project administration; writing—review and editing.

## Disclosure and competing interests statement

The authors declare that they have no conflict of interest.

## For more information

i   https://www.ahri.org/scientist/adrie-steyn/

ii  https://www.ebi.ac.uk/biostudies/studies/S-BIAD516

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
