## [Review Process File · EMBO Molecular Medicine]

A high-resolution 3D atlas of the spectrum of tuberculous and COVID-19 lung lesions

Gordon Wells, Joel Glasgow, Kievershen Nargan, Kapongo Lumamba, Madansein Rajhmun, Kameel Maharaj, Leon Perumal, Malcolm Matthew, Robert Hunter, Hayden Pacl, Jacelyn Peabody Lever, Denisie Stanford, Satinder Singh, Prachi Bajpai, Upendar Manne, Paul Benson, Steven Rowe, Stephan le Roux, Alex Sigal, Muofhe Tshibalanganda, Carlyn Wells, Anton Plessis, Mpumelelo Msimang, Threnesan Naidoo, and Adrie Steyn

DOI: 10.15252/emmm.202216283

Corresponding author: Adrie Steyn (asteyn@uab.edu)

Review Timeline:

Submission Date:	11th May 22
Editorial Decision:	5th Jul 22
Revision Received:	1st Sep 22
Editorial Decision:	13th Sep 22
Revision Received:	26th Sep 22
Accepted:	27th Sep 22

Editor: Zeljko Durdevic

Transaction Report:

5th Jul 2022

Dear Prof. Steyn,

Thank you for the submission of your manuscript to EMBO Molecular Medicine. Please accept my apologies for the delay in getting back to you due to the withdrawal of the referee #1 from the peer-review process and securing additional referees to evaluate your manuscript. We have now received feedback from the three reviewers who agreed to evaluate your manuscript. As you will see from the reports below, the referees acknowledge the interest of the study but also raise important concerns that should be addressed in a major revision.

We would welcome the submission of a revised version within three months for further consideration. Please let us know if you require longer to complete the revision.

Please use this link to login to the manuscript system and submit your revision: Link Not Available

I look forward to receiving your revised manuscript.

Yours sincerely,

Zeljko Durdevic

***** Reviewer's comments *****

Referee #2 (Remarks for Author):

Major comments

This study clarified the macro- and microscopic features of the spectrum of TB and COVID-19 lesions using μ /nCT. This study will fill the gap between HRCT images and pathological findings *ex vivo*. However, there are some concerns in this manuscript. The purpose of this study is not clear. Although we evaluate this study in terms of matching the findings of μ /nCT with the pathological findings, little consideration was given to pathological disregarded structures. In addition, μ /nCT findings in mycetoma cannot provide the information other than calcification and hemorrhage. Therefore, the advantage of μ /nCT is unclear compared with standard histopathology. Moreover, since μ /nCT cannot be used for patients without surgical resection or post-mortem, its clinical application is limited for diagnostic purposes. Although CT findings complement pathological findings in this study, it is necessary to show that there are stronger advantages of μ /nCT over pathological findings.

Minor comments

1. Line 311. We cannot confirm Figure 8C, D.
2. Figure S15C. Similar to COVID-19, μ CT appears to indicate the dilated vascularization close to the pleural surface. Are there any pathological findings such as occlusion or microangiopathy?

Referee #3 (Comments on Novelty/Model System for Author):

The technical quality of this paper is amongst the best that I have seen. The novelty is sufficient in that it reaffirms with high quality evidence what the field had recently realised. Ultimately this is an atlas, with technical superiority as the most important element. Thus this has medical impact higher than novelty alone. This is actual human tissue work, of course of a subset of severer than usual patients, but that is still far more reliable for human disease than model systems.

Referee #3 (Remarks for Author):

The work done by the authors is of the highest technical standard, showing applications of imaging tech that break current boundaries. The insights are useful, if not radically new, but importantly very convincing and thus clinically important by being definitive where previous evidences may have not been as convincing. The vascular abnormalities in SARS CoV2 would be such an area. Some suggestions are below

- 1) It may be useful to more clearly indicate what the atlas represents and what it doesn't, because of the limited choices of human tissue e.g. a mild infection that responds well to therapy (tuberculosis) or spontaneous resolution (SARS CoV2) is unlikely to be represented in the tissue sets described.
- 2) In the same context, it could be discussed what are the remaining gaps in knowledge, based on less definitive macro-imaging, worth pursuing further via micro-imaging.

Referee #4 (Remarks for Author):

The authors describe an interesting approach to using CT technologies to gain a better insight into structures and pathological processes in large parts or full tissues. Overall the manuscript is well written and plenty of figures supporting their points are presented. A few points should be clarified to improve the readability of the manuscript in particular for non-image technology specialists:

- 1) In the introduction there is no clear explanation of the basics of the CT technologies introduced. Please add a very brief description.
- 2) The difference between working with contrast agents or not is also not explained. Adding this would make the information presented in the manuscript easier to understand, particularly line 184-185 where contrast stained and non-contrast stained is being compared but it is unclear when which option would be used.
- 3) The technology is described as non-destructive but in figure 6 tissue is sliced, why?
- 4) In paraffin blocks, the main identifiable structures mentioned are calcium deposits, which according to lines 371-374 is important, but the point being made here is not clear. Please clarify. Now it seems that the point being made is that calcifications are present in TB, but also in latent TB, and that in latent TB it points to other diseases. This does not make sense to me.
- 5) Is deparaffination of tissue reversible?
- 6) In line 187 it seems that the references to figure 2D (nCT) and 2E (mCT) have been swapped compared to the legend 2D (mCT) and 2E (nCT). The legend seems to be correct and aligning with the conclusion in the text, please check this.
- 7) line 192, should the figure referred to be 2F rather than 2E?
- 8) When referring to different structures in figure 3E please refer to the letters of those structures by their letters to make it easier to find as done with C2 in line 214, but not for the blood lakes or first mention of the cavities in line 207 & 219
- 9) Please check figure referencing in line 306-313, figure 8 and 12 are referred to but don't exist, perhaps these should be referring to the supplementary figures?
- 10) In figure 1F, what are the white regions that are not visible as necrotic or calcified regions in fig 1E? Are they similar lesions in different depths of the tissue or is this some form of background that makes it hard to distinguish positives?

If these points are addressed and clarified the paper is suitable for publication.

Referee #2 (Remarks for Author):**Major comments**

This study clarified the macro- and microscopic features of the spectrum of TB and COVID-19 lesions using μ/n CT. This study will fill the gap between HRCT images and pathological findings ex vivo. However, there are some concerns in this manuscript. The purpose of this study is not clear. Although we evaluate this study in terms of matching the findings of μ/n CT with the pathological findings, little consideration was given to pathological disregarded structures. In addition, μ/n CT findings in mycetoma cannot provide the information other than calcification and hemorrhage. Therefore, the advantage of μ/n CT is unclear compared with standard histopathology. Moreover, since μ/n CT cannot be used for patients without surgical resection or post-mortem, its clinical application is limited for diagnostic purposes. Although CT findings complement pathological findings in this study, it is necessary to show that there are stronger advantages of μ/n CT over pathological findings.

The purpose of this study is not clear.

Response: We regret the lack of clarity; the purpose of this study is now clearly stated in the Introduction of the revised manuscript (**lines 121-123**).

Although we evaluate this study in terms of matching the findings of μ/n CT with the pathological findings, little consideration was given to pathological disregarded structures.

Response: To the best of our knowledge, this study is the first to examine cryptic, disregarded pathological structures in detail, which is *extremely* time consuming. While we did not elaborate on all disregarded structures present in these datasets, we believe that we have convincingly demonstrated the ability of μ CT to characterize unattributed structures within necrotic lesions (**Figure 6**, revised manuscript). Our primary goal is to demonstrate this as a key advantage of μ CT over histopathology, which represented a significant technical advance.

In addition, μ/n CT findings in mycetoma cannot provide the information other than calcification and hemorrhage. Therefore, the advantage of μ/n CT is unclear compared with standard histopathology.

Moreover, since μ/n CT cannot be used for patients without surgical resection or post-mortem, its clinical application is limited for diagnostic purposes. Although CT findings complement pathological findings in this study, it is necessary to show that there are stronger advantages of μ/n CT over pathological findings

Response: Our study highlights several advantages of μ CT compared with standard histopathology to examine mycetomas. These advantages (described in detail in **lines 345-373** of the revised manuscript) include, but are not limited to: (i) the non-destructive imaging which is impossible to achieve with traditional histology, (ii) 3D reconstruction of mycetoma structure and calcium crystals, which is not possible with conventional histology, and (iii) quantitation of biological features such as calcium crystal size, surface area, volume, quantity, and mycetoma and cavity volumes, which is impossible with conventional histopathology. Also, we reiterate key advantages of μ CT in the final paragraph of the Discussion (**lines 415-423** in the revised manuscript), specifically the scanning of FFPE libraries to expand the atlas of TB lung lesions and discovery of imaging biomarkers. Of course, we agree with the Reviewer about the (direct) limited clinical application.

Importantly, to further demonstrate the advantages of μ CT in examining FFPE human TB tissue, we scanned an additional sample and provide compelling evidence that direct visualization of necrotic lesions in wax blocks is possible without contrast staining. A surprising finding was that several of these apparent necrotic lesions are also partially calcified (and hence, easily detectable using μ CT), which is difficult to appraise through conventional histopathology. These **new data** are included in the revised manuscript (**Figure 5** and **lines 231-237** in the revised manuscript).

Minor comments

1. Line 311. We cannot confirm Figure 8C, D.

Response: The Reviewer is correct that the original manuscript referred to **Figure 8C, D** when the correct figure was **Figure S12**. This has error been corrected in the revised manuscript (**line 298**).

2. Figure S15C. Similar to COVID-19, μ CT appears to indicate the dilated vascularization close to the pleural surface. Are there any pathological findings such as occlusion or microangiopathy?

Response: Although this is a “control” (COVID-19/TB negative) lung, these lungs are not completely healthy as the decedents suffered from hepatocellular carcinoma, NASH cirrhosis, COPD, esophageal varicies, septic shock and anasarca etc. hence, the COVID-19 negative decedents did not show prominent pathology (e.g., microangiopathy etc) consistent with that of COVID-19 as the cause of death.

Referee #3 (Comments on Novelty/Model System for Author):

The technical quality of this paper is amongst the best that I have seen. The novelty is sufficient in that it reaffirms with high quality evidence what the field had recently realised. Ultimately this is an atlas, with technical superiority as the most important element. Thus, this has medical impact higher than novelty alone. This is actual human tissue work, of course of a subset of severer than usual patients, but that is still far more reliable for human disease than model systems.

Referee #3 (Remarks for Author):

The work done by the authors is of the highest technical standard, showing applications of imaging tech that break current boundaries. The insights are useful, if not radically new, but importantly very convincing and thus clinically important by being definitive where previous evidences may have not been as convincing. The vascular abnormalities in SARS CoV2 would be such an area. Some suggestions are below

- 1) It may be useful to more clearly indicate what the atlas represents and what it doesn't, because of the limited choices of human tissue e.g. a mild infection that responds well to therapy (tuberculosis) or spontaneous resolution (SARS CoV2) is unlikely to be represented in the tissue sets described.*
- 2) In the same context, it could be discussed what are the remaining gaps in knowledge, based on less definitive macro-imaging, worth pursuing further via micro-imaging.*

Response: We thank the Reviewer for the positive feedback and appreciate the suggestions to improve our manuscript. As requested, we have elaborated on what the atlas represents and what it does not, and how μ CT imaging can address remaining gaps in our knowledge. This has been described in the Discussion section in **lines 407-408** of the revised manuscript.

Referee #4 (Remarks for Author):

The authors describe an interesting approach to using CT technologies to gain a better insight into structures and pathological processes in large parts or full tissues. Overall the manuscript is well written and plenty of figures supporting their points are presented. A few points should be clarified to improve the readability of the manuscript in particular for non-image technology specialists:

- 1) In the introduction there is no clear explanation of the basics of the CT technologies introduced. Please add a very brief description.*

Response: We agree with the Reviewer and have added a brief description of μ CT technology in **lines 102-104** in the Introduction of the revised manuscript.

2) *The difference between working with contrast agents or not is also not explained. Adding this would make the information presented in the manuscript easier to understand, particularly line 184-185 where contrast stained and non-contrast stained is being compared but it is unclear when which option would be used.*

Response: We apologize for the incomplete description. A brief explanation of contrast agents was added to the Introduction of the revised manuscript (**lines 116-120**).

3) *The technology is described as non-destructive but in figure 6 tissue is sliced, why?*

Response: CT-based imaging is non-destructive. The reason for physically slicing the lung tissue was to optimize exposure times to the contrast agents. We had to determine the time it took for the contrast agent to penetrate tissue slices of a particular thickness. The result of this optimization process enabled us to examine a complete lower lobe in remarkable detail.

4) *In paraffin blocks, the main identifiable structures mentioned are calcium deposits, which according to lines 371-374 is important, but the point being made here is not clear. Please clarify. Now it seems that the point being made is that calcifications are present in TB, but also in latent TB, and that in latent TB it points to other diseases. This does not make sense to me.*

Response: We apologize for the confusion and have corrected the sentence in **lines 380-382** in the revised manuscript. The message that we were attempting to convey is that in addition to being a useful tool to examine active TB disease and healing (as is evident by calcification), this imaging tool has potential to help examine latent TB infection. For example, a large number of historical studies have shown that partially calcified lesions in decedents that died of causes other than TB contains viable, culturable and virulent *Mtb*. We have clarified this in **line 381** of the revised manuscript.

5) *Is deparaffination of tissue reversible?*

Response: Yes, deparaffination is reversible. We highlight this fact in **line 393** and in the Materials and Methods section of the revised manuscript.

6) *In line 187 it seems that the references to figure 2D (nCT) and 2E (mCT) have been swapped compared to the legend 2D (mCT) and 2E (nCT). The legend seems to be correct and aligning with the conclusion in the text, please check this.*

Response: The Reviewer is correct. We thank the Reviewer and have corrected this mistake in **line 175-176** of the revised manuscript.

7) *line 192, should the figure referred to be 2F rather than 2E?*

Response: The Reviewer is correct; we have corrected this error in **lines 180** of the revised manuscript.

8) *When referring to different structures in figure 3E please refer to the letters of those structures by their letters to make it easier to find as done with C2 in line 214, but not for the blood lakes or first mention of the cavities in line 207 & 219*

Response: As requested, we have added labels to the main text in **lines 194 and 204** of the revised manuscript.

9) *Please check figure referencing in line 306-313, figure 8 and 12 are referred to but don't exist, perhaps these should be referring to the supplementary figures?*

Response. We regret the error. This has been corrected in **lines 294-300** of the revised manuscript.

10) In figure 1F, what are the white regions that are not visible as necrotic or calcified regions in fig 1E? Are they similar lesions in different depths of the tissue or is this some form of background that makes it hard to distinguish positives?

Response: The Reviewer is correct that these regions are the buried portions of the same lesions within the tissue, which is below the surface (at different depths).

If these points are addressed and clarified the paper is suitable for publication.

Response: We thank the Reviewer for the helpful comments.

13th Sep 2022

Dear Prof. Steyn,

Thank you for the submission of your revised manuscript to EMBO Molecular Medicine. I am pleased to inform you that we will be able to accept your manuscript pending the following final amendments:

1) In the main manuscript file, please do the following:

- In M&M, please include statement that the experiments using human samples conformed to the principles set out in the WMA Declaration of Helsinki and the Department of Health and Human Services Belmont Report. Please also amend it in the "Author Checklist" as well.

- Please be aware that all datasets should be made freely available upon acceptance, without restriction. Please check "Author Guidelines" for more information. <https://www.embopress.org/page/journal/17574684/authorguide#availabilityofpublishedmaterial>

2) Source data: Please zip source data for Figure 4 and upload it as a single file.

3) The Paper Explained: Please add it to the main manuscript file.

4) Synopsis:

- Synopsis image: Please resize the image and upload it as a high-resolution jpeg file 550 px-wide x (250-400)-px high.

5) For more information: Please remove corresponding author's e-mail address. This space should be used to list relevant web links for further consultation by our readers. Could you identify some relevant ones and provide such information as well? Some examples are patient associations, relevant databases, OMIM/proteins/genes links, author's websites, etc...

6) Press release: Please inform us as soon as possible and latest at the time of submission of the revised manuscript if you plan a press release for your article so that our publisher could coordinate publication accordingly.

7) Please be aware that we use a unique publishing workflow for COVID-19 papers: a non-typeset PDF of the accepted manuscript is published as "Just Accepted" on our website. With respect to a possible press release, we have the option to not post the "Just Accepted" version if you prefer to wait with the press release for the typeset version. Please let us know whether you agree to publication of a "Just accepted" version or you prefer to wait for the typeset version.

8) As part of the EMBO Publications transparent editorial process initiative (see our Editorial at <http://embomolmed.embopress.org/content/2/9/329>), EMBO Molecular Medicine will publish online a Review Process File (RPF) to accompany accepted manuscripts. This file will be published in conjunction with your paper and will include the anonymous referee reports, your point-by-point response and all pertinent correspondence relating to the manuscript. Let us know whether you agree with the publication of the RPF and as here, if you want to remove or not any figures from it prior to publication. Please note that the Authors checklist will be published at the end of the RPF.

9) Please provide a point-by-point letter INCLUDING my comments as well as the reviewer's reports and your detailed responses (as Word file).

I look forward to reading a new revised version of your manuscript as soon as possible.

Yours sincerely,

Zeljko Durdevic

**** Reviewer's comments ****

Referee #2 (Remarks for Author):

This study evaluated the consistency of μ /nCT images and pathological findings. The author stated only objective facts and did not mention any newly discovered pathological findings. Hence, this content was not so innovative. Furthermore, it is a drawback that μ /nCT can be used only for surgical resection or post-mortem. Therefore, it cannot be used as a diagnostic modality in clinical situations. However, it is commendable that μ /nCT revealed disregarded structures and internal structures of calcification that conventional pathological examinations could not diagnose. In light of the possibility that μ /nCT may contribute to elucidating the pathogenesis of TB and COVID-19.

Referee #3 (Remarks for Author):

My concerns have been adequately addressed. I have no further concerns

Referee #4 (Remarks for Author):

The manuscript has been much improved with the clarifications in the text for certain approaches and conclusions that were not clear previously. The method described is interesting and I support publication of the manuscript.

The authors performed the requested editorial changes.

We are pleased to inform you that your manuscript is accepted for publication and is now being sent to our publisher to be included in the next available issue of EMBO Molecular Medicine.